# Numerical Analysis of an Electroless Plating Problem in Gas–Liquid Two-Phase Flow

**Po-Yi Wu** [1,2], **Olivier Pironneau** [1,*], **Po-Shao Shih** [2] and **ChengHeng Robert Kao** [2]

1   Laboratoire Jacques-Louis Lions, Sorbonne University, 75005 Paris, France; f04527030@ntu.edu.tw
2   Department of Materials Science and Engineering, National Taiwan University, Taipei 10617, Taiwan; f08527055@ntu.edu.tw (P.-S.S.); crkao@ntu.edu.tw (C.R.K.)
*   Correspondence: olivier.pironneau@gmail.com

**Abstract:** Electroless plating in micro-channels is a rising technology in industry. In many electroless plating systems, hydrogen gas is generated during the process. A numerical simulation method is proposed and analyzed. At a micrometer scale, the motion of the gaseous phase must be addressed so that the plating works smoothly. Since the bubbles are generated randomly and everywhere, a volume-averaged, two-phase, two-velocity, one pressure-flow model is applied. This fluid system is coupled with a set of convection–diffusion equations for the chemicals subject to flux boundary conditions for electron balance. The moving boundary due to plating is considered. The Galerkin-characteristic finite element method is used for temporal and spatial discretizations; the well-posedness of the numerical scheme is proved. Numerical studies in two dimensions are performed to validate the model against earlier one-dimensional models and a dedicated experiment that has been set up to visualize the distribution of bubbles.

**Keywords:** electroless plating process; two-phase flow; finite element method; two-velocities averaged model





## 1. Introduction

Electroless plating is an industrial chemical process aimed at forming a thin film or layer on a base substrate by reducing complex metal cations in a liquid solution [1–3]. This technique has been widely applied in various industries. For instance, surface decoration, hard-wearing coating, manufacture of hard-disc drive, printed circuit boards, etc. [3,4]. It is well known that many electroless plating processes produce parasite hydrogen bubbles, such as electroless nickel and electroless copper systems [5–7]. There are several works on the simulation of electroless processes that study the convection or migration of chemical species under a single-phase flow (e.g., [8,9]). As far as we know, there is no computational work on bubble generation in an electroless process. In large-scale electroless plating processes, the gas generation will not be a serious issue, in general, however, for micro-scale plating, hydrogen bubbles may prevent electrolytes from going into the region needed to be plated. The existence of relatively large bubbles has been an important issue in the study of micro-fluids [10–12]. As experiments are difficult, there is a need for a reliable numerical simulation tool.

From a theoretical physics point of view, electroless processes are complex; the entire physical system participates to the plating. Therefore, for the simulation, we chose a system that includes a gas–liquid two-phase flow, chemical species transport, surface reaction, and moving boundary due to deposition.

### 1.1. The Modeling

Because bubbles and sprays are frequent in engineering design with fluids, numerous papers on the modeling and simulation of gas–liquid two-phase flows have been published such as [13–17]. Working models to compute gas–fluid flows can be sorted into two

classes: (i) phase field or level set models where the gas–liquid interface is traced [18–21]; (ii) averaged models [22–24]. Several reasons support our choice for an averaged model: (i) There are many bubbles and their generation seems random, we only know that there is a higher chance of gas generation occurring in regions of higher concentration of dissolved gas; (ii) even if the bubble generation can be well predicted, vast amounts of bubbles are generated in short moments and the computational cost for capturing each bubble is prohibitive; (iii) interfacial terms (e.g., terms caused by a phase change) can be easily estimated with an averaged model (see Appendix A).

Experimentally, the bubbles are seen to get stuck at unexpected regions of the microchannel. This indicates that the velocities of the two phases are quite different. To allow a disparity of motion between the liquid phase and gaseous phase, a two-velocity model is preferred. Two-velocity one pressure models have been used earlier either with the Navier–Stokes equations for both phases [25] or with the Navier–Stokes equations for the liquid phase and a potential flow equation for the gaseous phase [26]; yet, while mathematical analyses for one-velocity models of multiphase flows are numerous, few are those which deal with two-velocity models [27–29]. Moreover, we are not aware of a mathematical analysis of the two-velocity one-pressure model, namely the existence of solutions in a variational setting, stability and convergence of the numerical schemes. In the present study, both velocities (liquid and gaseous phase) are governed by the Navier–Stokes equations for incompressible fluids with single common pressure, interpreted as the Lagrange multiplier (see [30,31] for the incompressibility equation of the mixture). In addition, the mass conservation for the fluids is coupled with the chemical species transport. By using the saddle point theorem known as the LBB condition for the Navier–Stokes equations we are able to derive well-posedness and stability results to this more complex two-velocity one-pressure system.

A system of linear convection–diffusion equations, with source terms due to phase changes, is used to define the concentration profiles of the chemical species. We use the mixed potential theory (see for instance [32]) to model the reaction boundary condition describing the electroless process; it is a Robin boundary condition subject to electron balance constraints. We further consider the boundary motion induced by the chemical species deposition on the reaction surface. The net result is a set of coupled equations for a system that includes gas–liquid fluid motion, chemical species transport, and moving boundary to simulate the plating process. Note that, in absence of bubbles, the proposed model reduces to the usual single-phase model (i.e. neglecting the existence of gas) which is compatible with previous studies such as [33]. Note also that the potential flow assumption for the gas in [26] is, to some extent, a special case of the current model.

### 1.2. Discretization, Stability, and Convergence

For numerical simulations, the Galerkin-characteristic method [34] is applied for temporal discretization. The finite element method, of degree two for the velocities and one for pressure and concentrations, is used for spatial discretization. The well-posedness of the numerical scheme for the coupled system is proved, i.e. existence of solution, stability, and uniqueness of solution for the linear systems at each time step.

The computer code is written for two-dimensional problems using FreeFEM++ [35], a high level partial differential equations (PDE) solver like COMSOL [36]. Using invariance in one spatial direction, we can reproduce the one-dimensional numerical simulation of [8]. Then we compare the numerical results with a real-world experiment done by one of the authors for this purpose.

### 1.3. Experiment and Comparison

An electrolyte with copper ions and formaldehyde flows in micro-channel between two parallel glass sheets (Figure 1). One piece of the sheet is partially glued on a copper plate whose longer side coincides with an edge of the inlet. Electroless copper plating was conducted in a water tank controlled at 50 °C with in situ recording via stereomicroscope (charged coupled device 311 digital camera CCD).

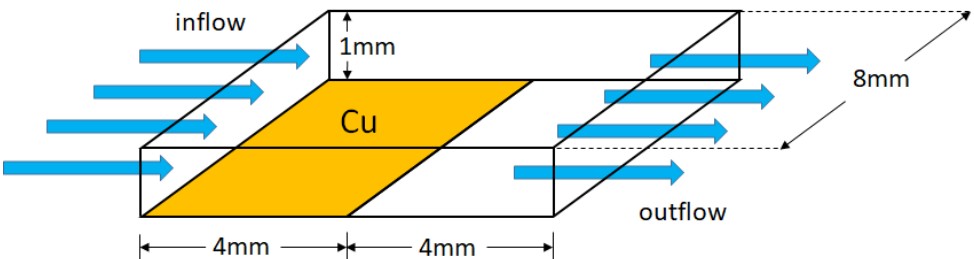

**Figure 1.** The geometry setting for both experiment and numerical simulation. Here, the yellow region indicates the copper plate glued on the sheet glass.

Results show that the bubbles are not only appearing on the copper plate, but also near the top glass sheet. In the video, one can see that there were several bubbles going to the top from the center or the bottom side of the channel. The bubble size distribution was not measured and it ought to be done in the future and combined with numerical simulations as in [37]. The measurement of the bubble densities in the experiment is possible by means of acoustic, optical, and laser diffraction approaches [38]. However, these techniques are hard (or expensive) to be adapted on a microfluidic scale.

The computer simulations qualitatively arrive at the same conclusion. The experiment indicates that the clustering of bubbles happens on both the top side and the bottom side of the channel. Second, the numerical simulation predicts that most bubbles are generated at an early stage and near the inlet. The experiment shows that the bubble generation is more exuberant near the inlet. This observation coincides with that of Figures 2 and 3a. The region near the inlet at $t = 20$ is of the highest concentration of dissolving hydrogen gas. In addition, large bubbles were observed at the back end of the copper plates, which is also the case numerically as seen in Figure 3b.

The comparison is qualitative but sufficient to assert that the simulation software is a more powerful tool to prototype future industrial designs. Potentially, it seems more reliable than experiments and it gives detailed information on the free boundary and on the speeds and concentrations of the chemical, which is highly important for the design of commercial systems. Thus, we are confident in the future of the numerical method. The mathematical properties of robustness, stability, and convergence, verified here numerically, are a certification of the computer software for future use.

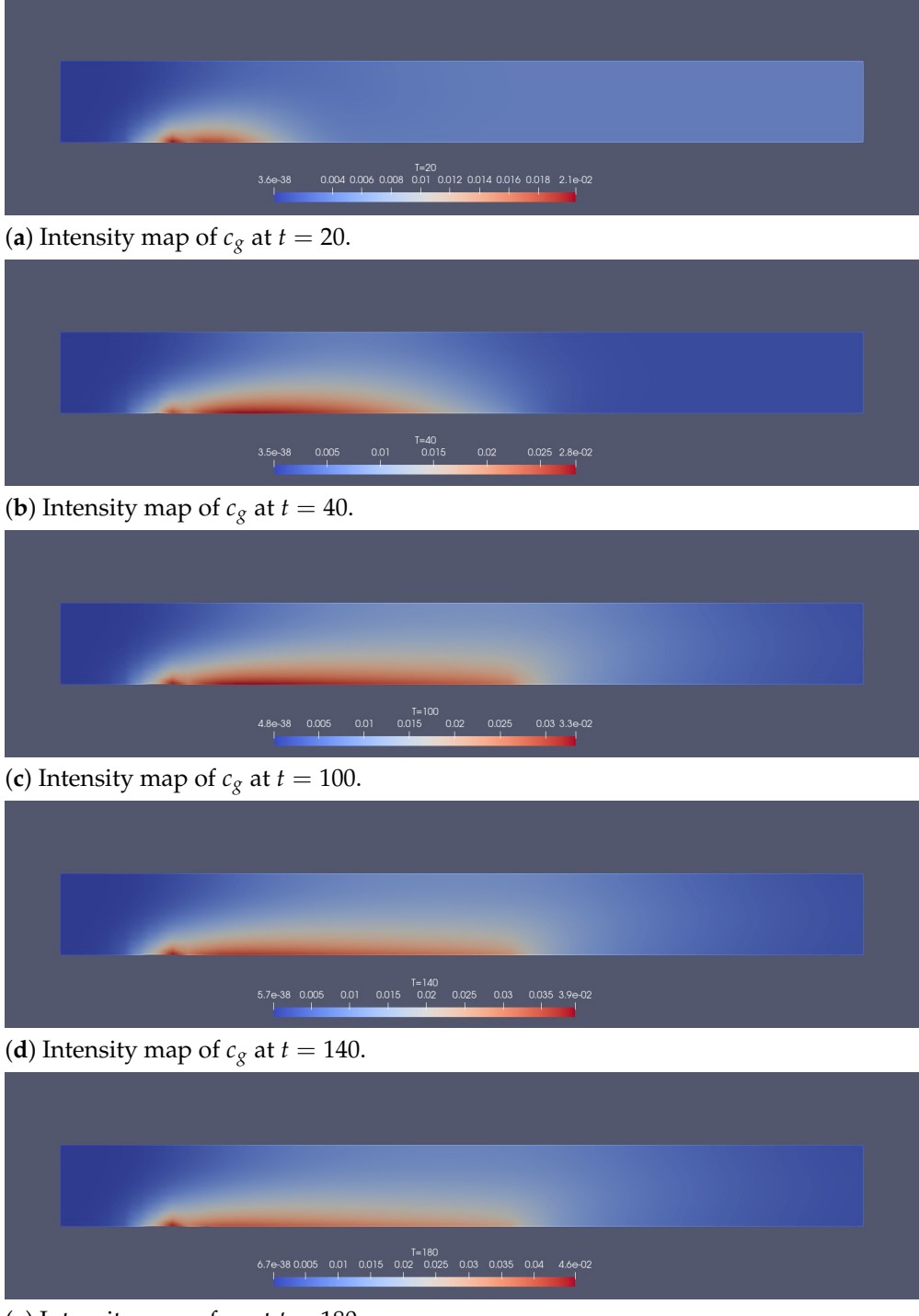

(**a**) Intensity map of $c_g$ at $t = 20$.

(**b**) Intensity map of $c_g$ at $t = 40$.

(**c**) Intensity map of $c_g$ at $t = 100$.

(**d**) Intensity map of $c_g$ at $t = 140$.

(**e**) Intensity map of $c_g$ at $t = 180$.

**Figure 2.** For Section 5.2: intensity maps of the concentration of dissolved gas. Notice that the gas seems to prefer to go up rather than to the right.

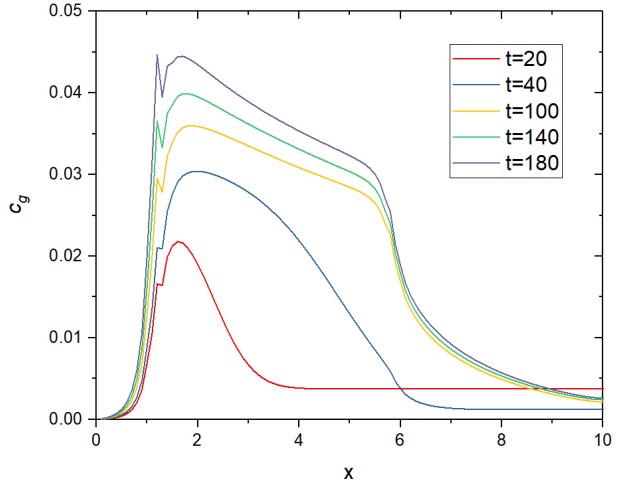

(**a**) For Section 5.2: Intensity map of $c_g$ on $S$ versus $x$.

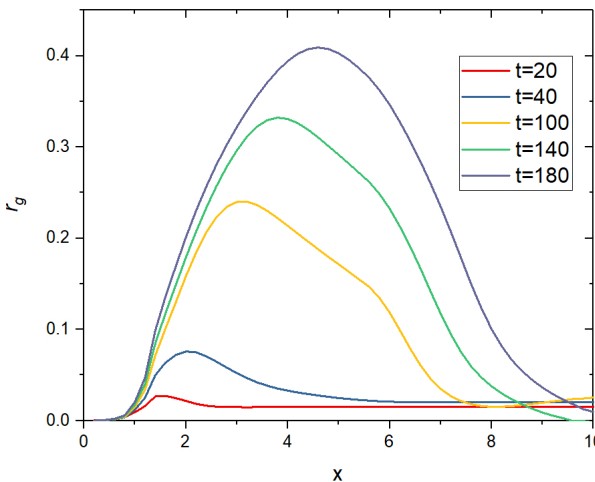

(**b**) For Section 5.2: Intensity map of $r_g$ on $S$ versus $x$.

**Figure 3.** Plots of $r_g$ and $c_g$ versus $x$ on the reaction surface $S$. The gas bubble density in the plating reaction zone can be observed.

## 2. Modeling Equations for Liquid-Gas Flow

Let $\Omega(t)$ be the time-dependent physical domain which is a thin channel between a top and a bottom plate. The boundary of $\Omega$ consists of the inlet $\Gamma_{in}$, the outlet $\Gamma_{out}$, the solid wall $\Gamma_{wall}$, and the reacting surface $S(t)$ (see Figure 4).

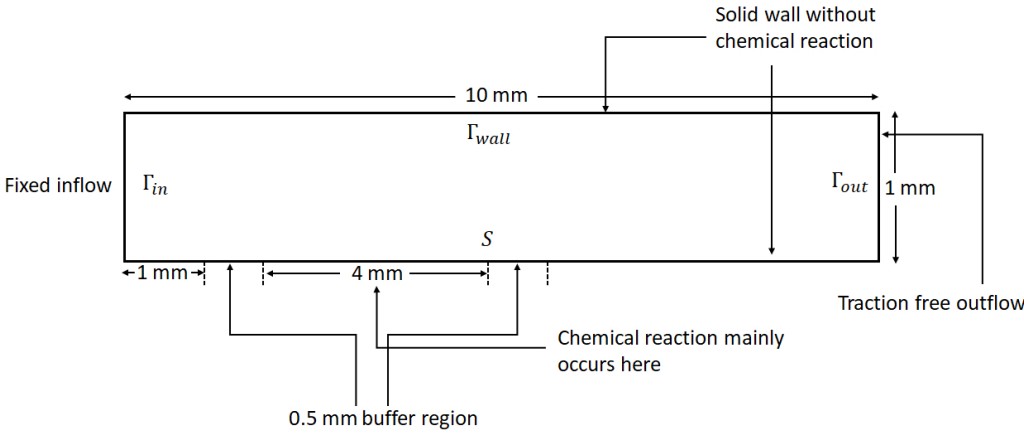

**Figure 4.** The computational domain $\Omega(t)$ for the test problem in Section 5.2 is initially a rectangle of size 10 mm $\times$ 1 mm. We assume a fixed inflow velocity and given chemical concentrations from the left on $\Gamma_{in}$, a solid wall on the top side with a no-slip condition for the velocity, and a traction-free outflow on $\Gamma_{out}$. On the bottom side, $S(t)$ is a free boundary and its motion is given by (23). However as the reaction site is active mostly for $x \in (1.5 \text{ mm}, 5.5 \text{ mm})$, we may block the chemical reactions for $x < 1$ mm to avoid a corner singularity at the entrance and also for $x > 6$ mm because experiments show that almost no plating occurs there. In the regions $x \in (1.0 \text{ mm}, 1.5 \text{ mm}) \cup (5.5 \text{ mm}, 6.0 \text{ mm})$ the numerical simulations may not be accurate due to the singularity caused by the discontinuity in the boundary conditions (see Figure 3 for details).

### 2.1. Volume Averaging

We review the derivation proposed by Ni and Beckermann [39].

Let $\Omega_0(x, t)$ be an small open set to be observed in $\Omega(t)$ and $\Omega_k \subset \Omega_0$ the set occupied by phase $k$ and bounded by the interface $\partial \Omega_k$. Assume that $\cup_k \Omega_k = \Omega_0$ and $\Omega_k \cap \Omega_j = 0$, $k \neq j$. Let $\boldsymbol{n}_k$ be a outer normal to $\partial \Omega_k$ and $\boldsymbol{w}_k$ the normal velocity of $\partial \Omega_k$.

Let $\Psi$ be a function of a slow variable $x$ and a fast variable $y$ due to the phase change. The volume average of $\Psi$ in phase $k$ is $\langle\Psi\rangle_k(x,t) = \frac{1}{V_0}\int_{\Omega_0(x,t)}\chi_k(y)\Psi(x,y)dy$, where $\chi_k$ is the indicator function of the domain of phase $k$ and $V_0 = \int_{\Omega_0}dx$, assumed constant. The intrinsic volume average is defined as

$$\langle\Psi\rangle_k^{(k)} = \frac{V_0}{V_k}\langle\Psi\rangle_k \text{ where } V_k = \int_{\Omega_0}\chi_k dy \tag{1}$$

The volume fraction $r_k = \frac{V_k}{V_0}$ has the properties $\sum_k r_k = 1$ and $\langle\Psi\rangle_k = r_k\langle\Psi\rangle_k^{(k)}$. Some useful averaging formulas are listed below [40,41]:

$$\left\langle\frac{\partial\Psi}{\partial t}\right\rangle_k = \frac{\partial\langle\Psi\rangle_k}{\partial t} - \frac{1}{V_0}\int_{\partial\Omega_k}\Psi_k w_k\cdot n_k dA, \qquad \langle\nabla\Psi\rangle_k = \nabla\langle\Psi\rangle_k + \frac{1}{V_0}\int_{\partial\Omega_k}\Psi_k n_k dA. \tag{2}$$

In principle, one should also introduce fast and slow time variables but it is assumed that spatially averaged functions are no longer varying fast in time.

*2.2. Mass Conservation*

We consider a gas and a liquid phase. Let $\rho_g$ be the density of gas, $\rho_l$ the density of liquid. We have the mass conservation for both phases ($l$ for liquid and $g$ for gas):

$$\partial_t(r_j\rho_j) + \nabla\cdot(r_j\rho_j u_j) = \dot{S}_j, \quad l = l,g \tag{3}$$

where $\dot{S}_g$ is the mass gained owing to the precipitation of dissolved gas, $\dot{S}_l$ is the mass loss when liquid evaporates into gas; $u_g(x,t)$, $u_l(x,t)$ are the volume averaged fluid flow of gas and liquid, respectively. Since the mass gained in gas balances the mass lost in liquid, we have

$$\dot{S}_g = -\dot{S}_l. \tag{4}$$

For chemical species, we assume that the ions are transported only by the liquid electrolyte. Let $c_s$ be the volume averaged concentration of metallic ions destined to be deposited on the reacting surface, $c_g$ the volume averaged concentration of dissolved gas, and $c_k, k = k_1,\ldots,k_M$ the volume averaged concentration of other chemical species participating to the chemical reactions. The equations for the concentrations are

$$\partial_t(r_l\rho_l c_j) + \nabla\cdot(r_l\rho_l c_j u_l) - \nabla\cdot(r_l\rho_l D_j\nabla c_j) - G_j = 0, \quad j = s,g,k. \tag{5}$$

where $G_j, j = s,k$ are interfacial terms due to the phase change. By (3), we can rewrite the above equation as

$$r_l\rho_l(\partial_t c_j + u_l\cdot\nabla c_j) - \nabla\cdot(r_l\rho_l D_j\nabla c_j) - G_j + \dot{S}_l c_j = 0, \quad j = s,g,k. \tag{6}$$

where $D_j$ are the diffusion coefficients. In particular, since the gas is consumed by the phase change, by assuming that the gas precipitation is linearly dependent on the dissolving gas concentration [42,43], we have

$$G_g = -\frac{1}{V_0}\int_{\partial\Omega_l}\rho_l c_g(u_l - w_l)\cdot n_l dA - \rho_l M_g Kr_l(c_g - c_{sat})^+. \tag{7}$$

In the above, $w_l$ is the interface velocity of $\partial\Omega_l$ and $K$ is a constant independent of $r_g$, $r_l$; $c_{sat}$ is the saturated concentration of the gas, $M_g$ is the reciprocal of the molar mass of the gas,

$$\dot{S}_g = Kr_l(c_g - c_{sat})^+. \tag{8}$$

Moreover, $G_j, j = s,k,g$ can be estimated by (see Appendix A)

$$G_j \approx \dot{S}_l c_j, \quad j = s,k, \quad G_g \approx \dot{S}_l c_g - \rho_l M_g Kr_l(c_g - c_{sat})^+. \tag{9}$$

For incompressible fluids, a volume conservation is derived from (3):

$$\sum_{\alpha=g,l} \frac{1}{\rho_\alpha} \left[ \partial_t(r_\alpha \rho_\alpha) + \nabla \cdot (r_\alpha \rho_\alpha \boldsymbol{u}_\alpha) - \dot{S}_\alpha \right] = 0. \tag{10}$$

By (4), the above reduces to

$$\nabla \cdot (r_g \boldsymbol{u}_g + r_l \boldsymbol{u}_l) = \dot{S}_g \left( \frac{1}{\rho_g} - \frac{1}{\rho_l} \right). \tag{11}$$

Recall that the physical domain is occupied either by gas or liquid, therefore $r_g(t) + r_l(t) = 1$ at all times.

### 2.3. Equations of Motion

Let $\mu_g$, $\mu_l$ be the viscosities of gas and liquid. The volume averaged Navier–Stokes equations are used for momentum balance (see [39]):

$$\partial_t(r_j \rho_j \boldsymbol{u}_j) + \nabla \cdot (r_j \rho_j \boldsymbol{u}_j \otimes \boldsymbol{u}_j) + r_j \nabla p_j - \mu_j \nabla \cdot (r_j D(\boldsymbol{u}_j)) + M_{D,j} = \boldsymbol{F}_j, \quad j = g, l \tag{12}$$

where $p_j$, $M_{D,j}$, $\boldsymbol{F}_j$, $j = l, g$ are pressure, drag and external force terms. Following [24,39]

$$M_{D,g} = C_D r_g |\boldsymbol{u}_g - \boldsymbol{u}_l|(\boldsymbol{u}_g - \boldsymbol{u}_l), \quad M_{D,l} = C_D r_g |\boldsymbol{u}_g - \boldsymbol{u}_l|(\boldsymbol{u}_l - \boldsymbol{u}_g), \tag{13}$$

where $C_D$ is a drag coefficient, and the external force are in fact the interfacial terms $\boldsymbol{F}_j = -\frac{1}{V_0} \int_{\partial \Omega_j} \rho_j \boldsymbol{u}_j (\boldsymbol{u}_j - \boldsymbol{w}_j) \cdot \boldsymbol{n}_j dA$, $j = l, g$, and $D(\boldsymbol{v}) = \nabla \boldsymbol{v} + (\nabla \boldsymbol{v})^T$ is twice the symmetric gradient of $\boldsymbol{v}$; $\boldsymbol{F}_g$ and $\boldsymbol{F}_l$ can be estimated by (see Appendix A)

$$\boldsymbol{F}_g \approx \dot{S}_g \boldsymbol{u}_g, \quad \boldsymbol{F}_l \approx \dot{S}_l \boldsymbol{u}_l, \tag{14}$$

In view of (11), following [44,45], we assume a constitutive relation $p = p_l = p_g$ in order to close the system of equations. The velocity fields of both phases are assumed to be 0 outside their own single phase region, respectively. Consequently, and by (3) and (12), the momentum equations simplify to

$$r_j \rho_j (\partial_t \boldsymbol{u}_j + (\boldsymbol{u}_j \cdot \nabla)\boldsymbol{u}_j) + r_j \nabla p - \mu_j \nabla \cdot (r_j D(\boldsymbol{u}_j)) + \gamma_j C_D r_g |\boldsymbol{u}_g - \boldsymbol{u}_l|(\boldsymbol{u}_g - \boldsymbol{u}_l) = 0,$$
$$j = g, l, \quad \text{with } \gamma_g = 1 \text{ and } \gamma_l = -1 \tag{15}$$

### 2.4. Boundary Conditions

We consider a fluid flow from an input boundary $\Gamma_{in}$ to an output boundary $\Gamma_{out}$ with a solid wall at the bottom, $\Gamma_{wall}$:

$$\boldsymbol{u}_j = \boldsymbol{u}_{in} \text{ on } \Gamma_{in}, \quad \boldsymbol{u}_j = 0 \text{ on } \Gamma_{wall}, \quad -\mu_j D(\boldsymbol{u}_j) \cdot \boldsymbol{n} + p\boldsymbol{n} = 0 \text{ on } \Gamma_{out}, \quad j = l, g. \tag{16}$$

The boundary conditions for $r_j$, $j = g, l$ are

$$r_g = \epsilon, \ r_l = 1 - \epsilon \text{ on } \Gamma_{in}, \quad \frac{\partial r_g}{\partial n} = \frac{\partial r_l}{\partial n} = 0 \text{ on } \partial\Omega \setminus \Gamma_{in}, \tag{17}$$

where $\epsilon$ is a fixed positive small constant.

The boundary conditions for the concentrations of chemicals are, with $c_{j,in}$ given:

$$c_j = c_{j,in} \text{ on } \Gamma_{in}, \quad \frac{\partial c_j}{\partial n} = 0 \text{ on } \Gamma_{wall} \cup \Gamma_{out}, \quad j = s, g, k \tag{18}$$

Referring to Figure 5, if $S(t)$ is the reaction surface, we denote $S_l(t) \subset S(t)$ the region occupied by the liquid and $S_g(t) := S(t) \setminus S_l(t)$ the region occupied by gas. Choos-

ing an arbitrary subset $W \subset S(t)$, the surface reaction takes place only on $W \cap S_l(t)$. Assuming that the concentration profile is uniform near the small region $W$, we have:

$-\int_W \rho_l D_j \frac{\partial c_j}{\partial n} dA = \int_{W \cap S_l(t)} \rho_l \frac{|I_j|}{z_j F} dA$. Therefore by dividing both sides by $\int_W 1 dA$:

$$-D_j \frac{\partial c_j}{\partial n} = \frac{|I_j|}{z_j F}, \quad j = s, k, \quad -D_g \frac{\partial c_g}{\partial n} = -\frac{\beta |I_s|}{z_s F} \tag{19}$$

for a positive number $\beta$ indicating the chemical equivalence for gaseous molecular generation; $F$ is the Faraday constant, and $z$ is the atomic number of the material. $I_j$ is the current density satisfying the Butler–Volmer equation

$$I_j = i_j(E_{mix})c_j^{\kappa_j} := L_j \left[ \exp\left( \frac{\alpha_j z_j F(E_{mix} - E_j)}{R\theta} \right) - \exp\left( \frac{-\beta_j z_j F(E_{mix} - E_j)}{R\theta} \right) \right] c_j^{\kappa_j}, \quad j = s, k, \tag{20}$$

where $R$ is the gas constant, $E_j$ are the chemical potentials of species $j$, $\theta$ is the temperature, $\alpha_j, \beta_j, L_j, \kappa_j$ are constants, and $E_{mix}$ is given by writing electrical neutrality :

$$I_s + \sum_k I_k = 0. \tag{21}$$

On $S(t)$, the fluid velocity induced by the deposition is

$$u_g = u_l = \frac{r_l V_s |I_s|}{z_s F} n. \tag{22}$$

where $V_s$ is a constant.

Finally, $S(t)$ moves according to

$$\dot{x}(t) = (u_g \cdot \mathbf{n})\mathbf{n}|_{x(t)}, \quad x(t) \in S(t) \tag{23}$$

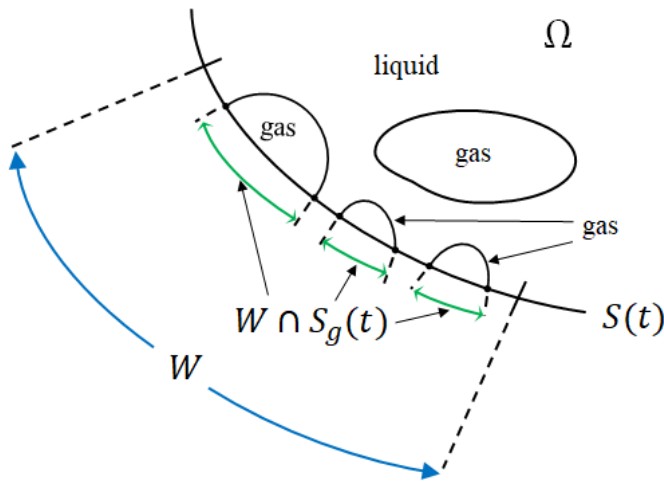

**Figure 5.** The reaction surface.

### 2.5. Single-Phase Flow

If there is no gaseous phase in the system and no dissolved gas in liquid, i.e., $r_g = c_g = 0$, then $u = u_l$ and $\dot{S}_g = 0$ and mass conservation reduces to $\nabla \cdot u = 0$. The convection–diffusion of chemicals become,

$$\partial_t c_j + u \cdot \nabla c_j - D_j \Delta c_j = 0, \quad j = s, k, \tag{24}$$

and the fluid system reduce to the Navier–Stokes equations:

$$\rho_l(\partial_t u + (u \cdot \nabla)u) - \mu_l \Delta u + \nabla p = 0, \qquad \nabla \cdot u = 0. \tag{25}$$

## 3. Numerical Method

### 3.1. Notations

If $f \in \mathbb{R}$, we denote by $f^+ := \max(f, 0)$ and by $f^- := -\min(f, 0)$. We denote by $\|\cdot\|_{L^p} := \|\cdot\|_{L^p(\Omega(t))}$ the $L^p$ norm on $\Omega(t)$, $\|\cdot\|_{W^{k,p}} := \|\cdot\|_{W^{k,p}(\Omega(t))}$ the $W^{k,p}$ norm, and $\|\cdot\|_{H^k} = \|\cdot\|_{W^{k,2}}$, $0 \le k \le +\infty$, $1 \le p \le +\infty$. Remembering that $c_k(x,t)$ is a vector, let us denote $C = (c_s, c_k^T, c_g)^T$.

We assume the densities $\rho_j$ constant and denote $\alpha_j = r_j \rho_j$ the mass fractions and $\nu_j = \mu_j / \rho_j$ the kinematic viscosities. The system is

$$\partial_t \alpha_j + u_j \cdot \nabla \alpha_j + \alpha_j \nabla \cdot u_j - \alpha_l \frac{\gamma_j K}{\rho_l}(c_g - c_{sat})^+ = 0, \quad j = l, g \tag{26}$$

$$\alpha_j \left( \partial_t u_j + u_j \cdot \nabla u_j + \rho_j^{-1} \nabla p \right) - \nu_j \nabla \cdot (\alpha_j D(u_j)) + \gamma_j C_D r_g |u_g - u_l|(u_g - u_l) = 0, \quad j = g, l \tag{27}$$

$$\alpha_l(\partial_t C + u_l \cdot \nabla C) - \nabla \cdot (\alpha_l D \cdot \nabla C) + (0, 0, M_g K \alpha_l (c_g - c_{sat})^+)^T = 0. \tag{28}$$

In addition as $\dot{S}_g = -\dot{S}_l = K\alpha_l(c_g - c_{sat})^+ / \rho_l$, we may use the redundant Equation (11):

$$\nabla \cdot (\frac{\alpha_g}{\rho_g} u_g + \frac{\alpha_l}{\rho_l} u_l) = \dot{S}_g \left( \frac{1}{\rho_g} - \frac{1}{\rho_l} \right). \tag{29}$$

### 3.2. Semi-Discrete Schemes

Let $T$ be the final time and $\delta t$ a time step. We denote by $\phi^m$, $m = 0, 1, \ldots, N := T/\delta t$, the numerical solution of any physical quantity $\phi$ at time $m\delta t$. Convection terms are approximated in time by the method of characteristics. Let $X_j^m(x) \approx x - u_j^m(x)\delta t$. Then

$$(\partial_t \alpha_j + u_j \cdot \nabla \alpha_j)|_{x, t=t^{m+1}} \approx \frac{1}{\delta t} \left( \alpha_j^{m+1}(x) - \alpha^m(X_j^m(x)) \right) := \frac{1}{\delta t} \left( \alpha_j^{m+1} - \alpha^m \circ X_j^m \right)|_x.$$

Consider the following scheme

$$\frac{1}{\delta t}(\alpha_l^{m+1} - \alpha_l^m \circ X_l^m) + \alpha_l^{m+1}\left( \nabla \cdot u_l^m + \frac{1}{\rho_l}K(c_g^m - c_{sat})^+ \right) = 0, \tag{30}$$

$$r_l^{m+1} = \alpha_l^{m+1}/\rho_l, \quad r_g^{m+1} = 1 - r_l^{m+1}, \quad \alpha_g^{m+1} = \rho_g r_g^{m+1} \tag{31}$$

$$\frac{1}{\delta t}\alpha_j^{m+1}(u_j^{m+1} - u_j^m \circ X_j^m) + \rho_j^{-1}\alpha_j^{m+1}\nabla p^{m+1} - \nu_j \nabla \cdot (\alpha_j^{m+1}D(u_j^{m+1}))$$
$$+ \gamma_j \rho_g^{-1} C_D \alpha_g^{m+1}|u_g^{m+1} - u_l^{m+1}|(u_g^{m+1} - u_l^{m+1}) = 0, \quad j = g, l, \tag{32}$$

$$\nabla \cdot (\rho_g^{-1}\alpha_g^{m+1}u_g^{m+1} + \rho_l^{-1}\alpha_l^{m+1}u_l^{m+1}) = \rho_l^{-1}K\alpha_l^{m+1}(c_g^{m+1} - c_{sat})^+ \left( \rho_g^{-1} - \rho_l^{-1} \right), \tag{33}$$

$$\frac{1}{\delta t}\alpha_l^{m+1}(C^{m+1} - C^m \circ X_j^m) - \nabla \cdot (\alpha_l^{m+1}D \cdot \nabla C^{m+1}) + (0, 0, M_g K \alpha_l^m (c_g^m - c_{sat})^+)^T = 0, \tag{34}$$

For electroless plating the domain is $\Omega^m = \{(x, y) : 0 < y < y^m(x), x \in (0, L)\}$, so it is updated by

$$y^{m+1}(x) = y^m(x) + \delta t u_{g2}^{m+1}(x), \quad x \in (0, L)$$

**Remark 1.** *Because of the asymmetrical treatment of $\alpha_g$ and $\alpha_l$ the scheme* (30) *does* not *imply that*

$$\frac{1}{\delta t}(\alpha_g^{m+1} - \alpha^m \circ X_g^m) + \alpha_g^{m+1}\nabla \cdot u_g^m = \rho_l^{-1}\alpha_l^{m+1}K(c_g^m - c_{sat})^+. \tag{35}$$

However, by (30), (31) and (33), we have

$$
\begin{aligned}
&\frac{1}{\delta t}(\alpha_g^{m+1} - \alpha_g^m \circ X_g^m) + \alpha_g^{m+1}\nabla \cdot \boldsymbol{u}_g^m + \frac{\rho_g}{\rho_l}(\alpha_l^{m+1} - \alpha_l^m)\nabla \cdot (\boldsymbol{u}_l^m - \boldsymbol{u}_g^m) \\
&+ \frac{1}{\delta t}(\alpha_g^m \circ X_g^m - \alpha_g^m \circ X_l^m) + (\boldsymbol{u}_g^m - \boldsymbol{u}_l^m) \cdot \nabla \alpha_g^m \\
&= \rho_l^{-1}\alpha_l^{m+1}K(c_g^m - c_{sat})^+ + \rho_l^{-1}(\frac{\rho_g}{\rho_l} - 1)(\alpha_l^{m+1} - \alpha_l^m)K(c_g^m - c_{sat})^+.
\end{aligned}
\tag{36}
$$

By a Taylor expansion at $x$, we obtain

$$
\begin{aligned}
\alpha_g^m(X_l^m(x)) - \alpha_g^m(X_g^m(x)) &= \delta t(\boldsymbol{u}_g^m - \boldsymbol{u}_l^m) \cdot \nabla \alpha_g^m(x) + O(\delta t^2), \\
\alpha_l^{m+1} - \alpha_l^m &= -\delta t(\boldsymbol{u}_l \cdot \nabla \alpha_l^m + \alpha_l^{m+1}\nabla \cdot \boldsymbol{u}_l^m) - \delta t K\alpha_l^{m+1}(c_g^m - c_{sat})^+ + O(\delta t^2).
\end{aligned}
\tag{37}
$$

Plugging (37) into (36), we have

$$
\frac{1}{\delta t}(\alpha_g^{m+1} - \alpha_g^m \circ X_g^m) + \alpha_g^{m+1}\nabla \cdot \boldsymbol{u}_g^m = \rho_l^{-1}\alpha_l^{m+1}K(c_g^m - c_{sat})^+ + O(\delta t).
\tag{38}
$$

So the scheme is consistent with the equation for $\alpha_g$.

### 3.3. Positivity

Positivity of $\alpha_l^{m+1}$ holds only if $\delta t$ is small enough. When positivity is required absolutely, an $O(\delta t)$ modification of (30) forces the positivity of $\alpha_l$:

$$
\begin{aligned}
\frac{1}{\delta t}\left(\alpha_l^{m+1}(x) - \alpha_l^m(X_l^m(x))\right) + \alpha_l^{m+1}\left(\nabla \cdot \boldsymbol{u}_l^m + \frac{1}{\rho_l}K(c_g^m - c_{sat})^+\right)^+ \\
= \alpha_l^m\left(\nabla \cdot \boldsymbol{u}_l^m + \frac{1}{\rho_l}K(c_g^m - c_{sat})^+\right)^-.
\end{aligned}
\tag{39}
$$

Indeed assume that $\alpha_l^m$ is strictly positive, or more precisely that $\alpha_l^m \geq \epsilon > 0$ for all $x$; then we have

$$
\begin{aligned}
\alpha_l^{m+1}\left[1 + \delta t(\nabla \cdot \boldsymbol{u}_l^m + \frac{1}{\rho_l}K(c_g^m - c_{sat})^+)^+\right] &= \alpha_l^m(X_l^m) + \delta t\alpha_l^m\left[\nabla \cdot \boldsymbol{u}_l^m + \frac{1}{\rho_l}K(c_g^m - c_{sat})^+\right]^- \\
&\geq \epsilon(1 + \delta t(\nabla \cdot \boldsymbol{u}_l^m + \frac{1}{\rho_l}K(c_g^m - c_{sat})^+)^-).
\end{aligned}
\tag{40}
$$

### 3.4. Stability

1. Let us show first that (30) generates a bounded sequence $\{\alpha_l^m\}_{m=1..N}$. For clarity we assume homogeneous data at the boundaries. With simplified notations

$$
\frac{1}{\delta t}(\alpha^{m+1} - \alpha^m \circ X^m) + \alpha^{m+1}(\nabla \cdot \boldsymbol{u}^m + \phi^m) = 0
$$

A multiplication by $\alpha^{m+1}$ and an integration on $\Omega^{m+1}$ leads to

$$
\|\alpha^{m+1}\|_{L^2}^2 = \int_{\Omega^{m+1}}\left[\alpha^{m+1}\left(\alpha^m \circ X^m - \delta t\left(\alpha^{m+1}\nabla \cdot \boldsymbol{u}^m + \phi^m\right)\right)\right]dx
$$

By the Cauchy Schwarz inequality and the positivity of $\phi^m$,

$$
\|\alpha^{m+1}\|_{L^2}^2 \leq \|\alpha^{m+1}\|_{L^2}\left(\int_{\Omega^{m+1}}\left[\alpha^m \circ X^m - \delta t\alpha^{m+1}\nabla \cdot \boldsymbol{u}^m\right]^2 dx\right)^{\frac{1}{2}}.
$$

The inverse of the determinant of the Jacobian of the transformation $x \mapsto X^m(x)$ is $1 + \delta t \nabla \cdot u + O(\delta t^2)$; therefore, for any smooth function $f$, in particular with $f = \alpha^m \circ X^m - \delta t \alpha^{m+1} \nabla \cdot u^m = \alpha^m \circ X^m (1 - \delta t \nabla \cdot u^m \circ X^m + O(\delta t^2))$,

$$\int_{\Omega^{m+1}} f^m \circ X^m = \int_{\Omega^m} f^m (1 + \delta t \nabla \cdot u + O(\delta t^2)) dx \Rightarrow \|\alpha^{m+1}\|_{L^2} \leq \|\alpha^m\|_{L^2}(1 + C\delta t^2).$$

where $C$ is a generic constant function of $\|\nabla^2 u^m\|_{L^\infty}$, the norm of the Hessian of $u$.

2. Stability of the scheme for $C$ is shown by the same argument.
3. Stability of the scheme for $u_g$ and $u_l$ is a consequence of a similar argument combined with the Ladhyzenskaya–Babuska–Brezzi saddle point theory (LBB) [46].

We denote by $(\cdot, \cdot)$ the $L^2$ inner product. For tensor-valued functions such that $f, g \in L^2(\Omega(t))^{mn}$, $m, n \in \mathbb{N}^+$, $(f, g) = \sum_{i=1}^{m} \sum_{j=1}^{n} (f_{ij}, g_{ij})$. With self explanatory notations, the equations for the velocities (32) and (33) are written in variational form as:

Find $u_g, u_l$ and $p$ satisfying the Dirichlet conditions and such that, $\forall \widehat{v}_g, \widehat{v}_l \in V_0^{m+1} := \left( H_0^1(\Omega^{m+1}) \right)^2$ and $\forall \widehat{q} \in P^{m+1} := L^2(\Omega^{m+1})/\mathbb{R}$,

$$\begin{aligned}
&(\beta_g u_g, \widehat{v}_g) + (\beta_l u_l, \widehat{v}_l) + \frac{1}{2}(\alpha_g D(u_g), D(\widehat{v}_g)) + \frac{1}{2}(\alpha_l D(u_l), D(\widehat{v}_l)) \\
&- \left( p, \nabla \cdot \left( \frac{\alpha_g}{\rho_g} \widehat{v}_g + \frac{\alpha_l}{\rho_l} \widehat{v}_l \right) \right) + \left( \widehat{q}, \nabla \cdot \left( \frac{\alpha_g}{\rho_g} u_g + \frac{\alpha_l}{\rho_l} u_l \right) \right) = (L_g, \widehat{v}_g) + (L_l, \widehat{v}_l) + (\widehat{q}, f).
\end{aligned} \tag{41}$$

where, for $j = g, l$, $\alpha_j := \alpha_j^{m+1}$, $\beta_j := \frac{1}{\delta t} \alpha_j + \frac{C_D}{\rho_g} \alpha_g |u_g^m - u_l^m|$,

$$L_j := \frac{1}{\delta t} \alpha_j u_l^m \circ X_j^m + \frac{C_D}{\rho_g} \alpha_g |u_g^m - u_l^m| u_{!j}^m, \quad f := \rho_l^{-1} K \alpha_l (c_g^{m+1} - c_{sat})^+ \left( \rho_g^{-1} - \rho_l^{-1} \right). \tag{42}$$

and where $!g = l$, $!l = g$. Here $H_0^1$ is the subspace of $H^1$ of functions which are zero on the Dirichlet boundaries.

Note that the above is a semi-linearization of (32) and (33). However in Algorithm 1 below, the nonlinear problem is solved by an iterative fixed point which uses (41) and (42).

The LBB theorem says that the solution of (41) exists and is unique because, for every $p \in P^{m+1}$ there is a (non-unique) $w \in V_0^{m+1}$ with

$$(\nabla \cdot w, \widehat{q}) = (f, \widehat{q}), \quad \forall \widehat{q} \in P^{m+1},$$

provided that $\int_{\Gamma_{in}} u_{in} \cdot n = \int_{\Omega^{m+1}} f dx$. Let us show stability in the special case $f = 0$ because one can always subtract $w$ from $\frac{\alpha_g}{\rho_g} u_g + \frac{\alpha_l}{\rho_l} u_l$ so as to work with $u_{g,in} = u_{l,in} = 0$ and $f = 0$.

Thus, setting $\widehat{v}_g = u_g$, $\widehat{v}_l = u_l$ and $\widehat{q} = p$ leads to

$$\frac{1}{\delta t}\|\sqrt{\alpha_g} u_g\|_{L^2}^2 + \frac{1}{2}\|\sqrt{\alpha_g} D(u_g)\|_{L^2}^2 + \frac{1}{\delta t}\|\sqrt{\alpha_l} u_l\|_{L^2}^2 + \frac{1}{2}\|\sqrt{\alpha_l} D(u_l)\|_{L^2}^2 \leq (L_g, u_g) + (L_l, u_l). \tag{43}$$

By the same argument used above, it implies that $u_j$, $j = g, l$ is bounded. Indeed, assuming $\alpha^m \geq 0$,

$$\begin{aligned}
\delta t (L_j, u_j) &= \int_{\Omega^{m+1}} \alpha_j u_j^m \circ X_j^m \cdot u_j dx \leq \left\| \sqrt{\alpha_j} u_j \right\|_{L^2} \left( \int_{\Omega^{m+1}} \alpha_j \left| u_j^m \circ X_j^m \right|^2 dx \right)^{\frac{1}{2}} \\
&= \left\| \sqrt{\alpha_j} u_j \right\|_{L^2} \left( \int_{\Omega^{m+1}} \left( \alpha_j^m \circ X_j^m - \alpha \delta t (\nabla \cdot u_l^m + \frac{1}{\rho_l} K(c_g^m - c_{sat})^+) \right) \left| u_j^m \circ X_j^m \right|^2 dx \right)^{\frac{1}{2}} \\
&\leq \left\| \sqrt{\alpha_j} u_j \right\|_{L^2} \left( \int_{\Omega^{m+1}} \left( \alpha_j^m \circ X_j^m - \alpha \delta t \nabla \cdot u_l^m \right) \left| u_j^m \circ X_j^m \right|^2 dx \right)^{\frac{1}{2}} \\
&\leq \left\| \sqrt{\alpha_j} u_j \right\|_{L^2} \left( \int_{\Omega^{m+1}} \alpha_j^m \circ X_j^m \left( 1 - \delta t \nabla \cdot u_l^m + O(\delta t^2) \right) \left| u_j^m \circ X_j^m \right|^2 dx \right)^{\frac{1}{2}}
\end{aligned}$$

$$= \left\| \sqrt{\alpha_j} \boldsymbol{u}_j \right\|_{L^2} \left( \int_{\Omega^m} \left( \alpha_j^m (1 + O(\delta t^2)) \right) \left| \boldsymbol{u}_j^m \right|^2 dx \right)^{\frac{1}{2}}$$

$$\leq \left\| \sqrt{\alpha_j} \boldsymbol{u}_j \right\|_{L^2} (1 + \delta t^2 C(\|c_g^m\|_{L^\infty}, \|\nabla^2 \boldsymbol{u}_l^m\|_{L^\infty})) \left\| \sqrt{\alpha_j^m} \boldsymbol{u}_j^m \right\|_{L^2} \quad (44)$$

for some generic constant $C$ depending on $\|c_g^m\|_{L^\infty}$ and $\|\nabla^2 \boldsymbol{u}_l^m\|_{L^\infty}$. Finally, we obtain

$$|||\boldsymbol{u}_g^{m+1}, \boldsymbol{u}_l^{m+1}|||_{m+1} \leq (1 + C\delta t^2)|||\boldsymbol{u}_g^m, \boldsymbol{u}_l^m|||_m + \delta t \frac{C_D}{\rho_g} \|\frac{\alpha_g}{\alpha_j^m}(\boldsymbol{u}_g^m - \boldsymbol{u}_l^m)\|_{L^\infty} |||\boldsymbol{u}_g^m, \boldsymbol{u}_l^m|||_m, \quad (45)$$

where

$$|||\boldsymbol{u}_g, \boldsymbol{u}_l|||_m^2 := \sum_{j=g,l} \left\| \sqrt{\alpha_{j,h}^m} \boldsymbol{u}_j \right\|_{L^2}^2 + \tfrac{1}{2}\delta t \left\| \sqrt{\alpha_{l,h}^m} D(\boldsymbol{u}_j) \right\|_{L^2}^2, \quad m = 0, \dots, N.$$

This estimate would be optimal if the constant $C$ did not depend on the Hessian of the velocities. This is a drawback of the characteristic method and of the unsophisticated treatment of the nonlinearity. At the expense of long mathematical arguments it could be fixed as in [34], the scheme would be $H^1$ stable.

4.   Note that we have swept under the rug the fact that at level $m$ the domain of definition of the functions is $\Omega^m$ and at level $m+1$ it is $\Omega^{m+1}$. The problem can be solved but at the cost of difficult notations and iterations between $y^{m+1}$ and $\boldsymbol{u}^{m+1}$; for details see [47].

## 4. Finite Element Implementation

For simplicity, the physical domain $\Omega(t)$ is assumed to be a two-dimensional polygonal domain.

### 4.1. Mesh

Let $\{\mathcal{K}_h(t)\}_{h>0}$ be an affine, shape regular (in the sense of Ciarlet [48]) family of mesh conforming to $\Omega(t)$. The conforming Lagrange finite element space of degree $p$ on $\Omega(t)$ is

$$X_{h,t}^p = \{v \in C^0(\Omega(t)) : v|_K \in P^p, \forall K \in \mathcal{K}_h(t)\}, \quad (46)$$

where $P^p$ is the space of polynomials of degree $p$ of $\mathbb{R}^2$.

Let $\{\phi_1, \dots \phi_{N_q}\}$ be the nodal Lagrange basis of $X_{h,t}^1$. If the vertices are denoted by $\{\boldsymbol{q}^i\}_1^{N_q}$, then $\phi_i(\boldsymbol{q}_j) = \delta_{ij}$. Let $S_i$ be the support of $\phi_i$ and let $S_{ij} := S_i \cap S_j$. If $E$ is a union of triangles, define $\mathcal{I}(E) := \{i \in \{1, \dots, N_q\} : |S_i \cap E| \neq 0\}$. Finally, the local minimum mesh size of $K \in \mathcal{K}_h(t)$ is $h_K(t) := 1/\max_{i \in \mathcal{I}(K)} \|\nabla \phi_i\|_{L^\infty(K)}$, and the global minimum mesh size is $h(t) := \min_{K \in \mathcal{K}_h} h_K(t)$.

We assume that the connectivity of the mesh $\mathcal{K}_h(t)$ never changes with time.

### 4.2. Spatial Discretization

We use the Hood–Taylor element: the velocities are in $V_h(t) := (X_{h,t}^2)^2$ and the pressure is in $P_h(t) := X_{h,t}^1$. For the volume fractions and the concentrations we use also $P_h(t)$.

Recall that the nodes of $X_{h,t}^2$ are the vertices and the middle of the edges. Denote by $\{\psi_1, \dots, \psi_{N_a}\}$ the nodal Lagrange basis of $X_{h,t}^2$. As before $\Omega^m := \Omega(t^m)$, $P_h^m = X_{h,t^m}^1$ and $V_h^m = (X_{h,t^m}^2)^2$.

On the boundaries where Dirichlet conditions are set, the functions are known. We denote $P_{0h}^m$ and $V_{0h}^m$, the corresponding spaces where basis functions attached to a Dirichlet node are removed.

### 4.2.1. Mass Fractions

Given $\alpha_{l,h}^m, c_{g,h}^m \in P_h^m$ and $\boldsymbol{u}_{l,h}^m \in V_h^m$, find $\alpha_{l,h}^{m+1} \in P_h^{m+1}$ satisfying the Dirichlet boundary conditions and such that

$$\frac{1}{\delta t}\left(\alpha_{l,h}^{m+1} - \alpha_{l,h}^m \circ X_{l,h}^m, \widehat{q}_h\right) + \left(\alpha_{l,h}^{m+1}(\nabla \cdot \boldsymbol{u}_{l,h}^m + \rho_l^{-1}K(c_{g,h}^m - c_{sat})^+), \widehat{q}_h\right) = 0, \ \forall \widehat{q}_h \in P_{0h}^{m+1}, \quad (47)$$

where $X_{j,h}^m(x) = x - \delta t\boldsymbol{u}_{j,h}^m(x)$ for $x \in \Omega^m$, $j = g, l$. Then we let $\alpha_{g,h}^{m+1} = \rho_g(1 - \rho_l^{-1}\alpha_{l,h}^{m+1})$.

**Remark 2.** A modification similar to (39) will insure the positivity of $\alpha_{l,h}^{m+1}$.

### 4.2.2. Concentration Profiles

Given $\alpha_{l,h}^{m+1} \in P_h^{m+1}, \alpha_{l,h}^m \in P_h^m, \boldsymbol{u}_{l,h}^m \in V_h^m, C_h^m \in (P_h^{m+1})^{2+k_M}$, find $C_h^{m+1} \in (P_h^{m+1})^{2+k_M}$ such that

$$\frac{1}{\delta t}\left(\alpha_{l,h}^{m+1}(C_h^{m+1} - C_h^m \circ X_l^m), \widehat{w}_h\right) + \left(\alpha_{l,h}^{m+1}D\nabla C_h^{m+1}, \nabla \widehat{w}_h\right) + \left(M_g K\alpha_{l,h}^m(c_{g,h}^m - c_{sat})^+, \widehat{w}_{g,h}\right)$$
$$+ \left(I(E_{mix,h}^{m+1})(C_h^{m+1})^\kappa, \widehat{w}_h\right)_{L^2(S(t^{m+1}))} = 0 \ \forall \widehat{w}_h \in (P_{0h}^{m+1})^{2+k_M}, \quad (48)$$

subject to

$$\sum_{j=s,k} i_j(E_{mix,h}^{m+1})(c_{j,h}^{m+1})^{\kappa_j}(\boldsymbol{q}_i) = 0, \ \text{ for each nodal point } \boldsymbol{q}_i \text{ on } S^m \quad (49)$$

where $\widehat{w}_{g,h}$ is the last component of $\widehat{w}_h$ and

$$I(E_{mix,h}^{m+1}) = \text{diag}\left(\frac{|i_s(E_{mix,h}^{m+1})|}{z_sF}, \frac{|i_k(E_{mix,h}^{m+1})|}{z_kF}, -\frac{\beta|i_s(E_{mix,h}^{m+1})|}{z_sF}\right),$$
$$(C_h^{m+1})^\kappa = \left((c_{s,h}^{m+1})^{\kappa_s}, (c_{k,h}^{m+1})^{\kappa_k}, (c_{s,h}^{m+1})^{\kappa_s}\right)^T$$

for $i_s, i_k$ defined by (20).

### 4.2.3. Two-Phase Flow

Given $\alpha_{j,h}^{m+1} \in P_h^{m+1}, j = g, l, c_{g,h}^{m+1} \in P_h^{m+1}$, and $\boldsymbol{u}_{j,h}^m \in V_h^m$, find $\boldsymbol{u}_{j,h}^{m+1} \in V_h^{m+1}, j = g, l$ and $p_h^{m+1} \in P_h^{m+1}/\mathbb{R}$ such that

$$\sum_{j=g,l}\left\{\frac{1}{\delta t}\left(\alpha_{j,h}^{m+1}(\boldsymbol{u}_{j,h}^{m+1} - \boldsymbol{u}_j^m \circ X_{j,h}^m), \widehat{\boldsymbol{v}}_{j,h}\right) + \frac{1}{2}\nu_j\left(\alpha_j^{m+1}D(\boldsymbol{u}_{j,h}^{m+1}), D(\boldsymbol{v}_{j,h})\right)\right.$$
$$\left. + \gamma_j\rho_g^{-1}C_D\left(\alpha_{g,h}^{m+1}|\boldsymbol{u}_{g,h}^{m+1} - \boldsymbol{u}_{l,h}^{m+1}|(\boldsymbol{u}_{g,h}^{m+1} - \boldsymbol{u}_{l,h}^{m+1}), \widehat{\boldsymbol{v}}_{j,h}\right) - \left(p_h^{m+1}, \nabla \cdot (\rho_j^{-1}\alpha_{j,h}^{m+1}\widehat{\boldsymbol{v}}_{j,h})\right)\right\} = 0 \quad (50)$$

$$\left(\widehat{q}_h, \nabla \cdot (\rho_g^{-1}\alpha_{g,h}^{m+1}\boldsymbol{u}_{g,h}^{m+1} + \rho_l^{-1}\alpha_{l,h}^{m+1}\boldsymbol{u}_{l,h}^{m+1})\right) = \left(\widehat{q}_h, \frac{K}{\rho_l}\alpha_{l,h}^{m+1}(c_{g,h}^{m+1} - c_{sat})^+\left(\rho_g^{-1} - \rho_l^{-1}\right)\right) \quad (51)$$

for all $\widehat{\boldsymbol{v}}_{j,h} \in V_{0h}^{m+1}, j = g, l$ and $\widehat{q}_h \in P_h^{m+1}/\mathbb{R}$.

### *4.3. Fixed Point Iterative Solution of (50) and (51)*

The system (50) and (51) is nonlinear. Algorithm 1 is proposed.

---

**Algorithm 1:** A semi-lineariazation for solving (50) and (51).

---

Let $L_g$, $L_l$, $f$ be defined by (42).

**Data:** Set $u_j = u_{j,h}^m$, $j = g, l$.

**for** $n = 1 \ldots N$ **do**

> Set $\beta_j = \left( \frac{1}{\delta t} \alpha_{j,h}^{m+1} + \frac{C_D}{\rho_g} \alpha_{g,h}^{m+1} |u_g - u_l| \right)$,
>
> Find $u_g, u_l$ and $p$ sastifying the Dirichlet conditions and such that, $\forall \widehat{v}_g, \widehat{v}_l \in V_{0h}^{m+1}$ and $\forall \widehat{q} \in P_h^{m+1}/\mathbb{R}$
>
> $$
> \begin{aligned}
> & \left( \beta_g u_g, \widehat{v}_g \right) + \left( \beta_l u_l, \widehat{v}_l \right) + \frac{1}{2} \left( \alpha_{g,h}^{m+1} D(u_g), D(\widehat{v}_g) \right) + \frac{1}{2} \left( \alpha_{l,h}^{m+1} D(u_l), D(\widehat{v}_l) \right) \\
> & - \left( p, \nabla \cdot \left( \frac{\alpha_{g,h}^{m+1}}{\rho_g} \widehat{v}_g + \frac{\alpha_{l,h}^{m+1}}{\rho_l} \widehat{v}_l \right) \right) + \left( \widehat{q}, \nabla \cdot \left( \frac{\alpha_{g,h}^{m+1}}{\rho_g} u_g + \frac{\alpha_{l,h}^{m+1}}{\rho_l} u_l \right) \right) \\
> & = \left( L_g, \widehat{v}_g \right) + \left( L_l, \widehat{v}_l \right) + \left( \widehat{q}, f \right).
> \end{aligned}
> \tag{52}
> $$

**end**

Set $u_j^{n+1} = u_j$, $j = g, l$.

---

### 4.4. Consistence and Stability

Variational formulations discretized by finite element methods inherit the stability and consistency of the continuous equations. The LBB theorem applies also to the Hood–Taylor element for velocity pressure problems. Therefore, as in the continuous case, the $H^1$ norms of $\alpha_j^{m+1}, u_j^{m+1}, C_j^{m+1}$ are less than $(1 + C(\|\nabla^2 u_l^m\|_{L^\infty})\delta t)$ times the $H^1$ norms of $\alpha_j^m, u_j^m, C_j^m$. If we could show that $C(\ )$ is bounded, then it would imply that the scheme converges when $\delta t \to 0$.

### 4.5. Solvability of the Linear System in Matrix Form

Let $\boldsymbol{\zeta} = (\zeta_g, \zeta_l) \in (V_h^{m+1})^2$, $\alpha_{j,h}^{m+1} \in P_h^{m+1}$, $\alpha_{j,h} \geq \epsilon$ for some constant $\epsilon > 0$, $j = g, l$.

To study the solvability of (50)–(51), we consider a simpler case with $u_g = u_l = 0$ on $\partial \Omega^{m+1} \setminus \Gamma_{out}$, and take the linearized approximation on the drag force terms. The problem reads: Find $u_h^{m+1} := (u_{g,h}^{m+1}, u_{l,h}^{m+1}) \in (V_{0h}^{m+1})^2$ and $p_h^{m+1} \in P_h^{m+1}/\mathbb{R}$ satisfying

$$
a_{\boldsymbol{\zeta}}(u_h^{m+1}, \widehat{v}_h) + b(p_h^{m+1}, \widehat{v}_h) = F(\widehat{v}_h), \quad b(\widehat{q}_h, u_h^{m+1}) = G(\widehat{q}_h),
\tag{53}
$$

where, for $u = (u_g, u_l), v = (v_g, v_l) \in (V_{0h}^{m+1})^2, q \in P_h^{m+1}$,

$$
\begin{aligned}
a_{\boldsymbol{\zeta}}(u, v) &= \sum_{j=g,l} \left[ \frac{1}{\delta t} \left( \alpha_{j,h}^{m+1} u_j, v_j \right) + \frac{1}{2} \nu_j \left( \alpha_{j,h}^{m+1} D(u_j), D(v_j) \right) \right] \\
& \quad + \rho_g^{-1} C_D \left( \alpha_{g,h}^{m+1} |\zeta_g - \zeta_l| (u_g - u_l), v_g - v_l \right) \\
b(q, v) &= - \left( q, \nabla \cdot (\rho_g^{-1} \alpha_{g,h}^{m+1} v_g + \rho_l^{-1} \alpha_{l,h}^{m+1} v_l) \right) \\
F(v) &= \sum_{j=l,g} \frac{1}{\delta t} \left( \alpha_{j,h}^{m+1} u_{j,h}^m (X_{j,h}^m(x)), v_j \right) \quad G(q) = \left( q, \rho_l^{-1} K \alpha_{l,h}^{m+1} (c_{g,h}^{m+1} - c_{sat})^+ (\rho_g^{-1} - \rho_l^{-1}) \right).
\end{aligned}
\tag{54}
$$

On the basis of $V_h^{m+1}$ and $P_h^{m+1}$, we can write

$$
u_{g,h}^{m+1} = \sum_{i=1}^{2N_a} u_{g,i}^{m+1} \boldsymbol{\psi}_i, \quad u_{l,h}^{m+1} = \sum_{i=1}^{2N_a} u_{l,i}^{m+1} \boldsymbol{\psi}_i, \quad p_h^{m+1} = \sum_{i=1}^{N_q} p_i^{m+1} \phi_i,
\tag{55}
$$

More precisely $\{\boldsymbol{\psi}_1, \ldots, \boldsymbol{\psi}_{2N_a}\}$ is $\{\psi_1 e_1, \ldots, \psi_{N_a} e_1, \psi_1 e_2, \ldots, \psi_{N_a} e_2\}$ for $e_1 = (1, 0)^T$ and $e_2 = (0, 1)^T$.

Problem (53) can be formally expressed as a system of linear equations:

$$
\Phi U^{m+1} = F^m,
\tag{56}
$$

where $\boldsymbol{\Phi}$ is a $(4N_a + N_q) \times (4N_a + N_q)$ matrix, $\boldsymbol{U}^{m+1}$ and $\boldsymbol{F}^m$ are $(4N_a + N_q)$ vectors. Note that $\boldsymbol{\Phi}$ has the form

$$\boldsymbol{\Phi} = \begin{pmatrix} A & B \\ B^T & O \end{pmatrix}, \text{ with } A = \begin{pmatrix} A_g & A_{mix} \\ A_{mix} & A_l \end{pmatrix}. \tag{57}$$

with, for $i, j = 1, \ldots, 2N_a$, $n = j = 1, \ldots, N_q$, $k = g, l$,

$$A_{k_{ij}} = \frac{1}{\delta t} \left( \alpha_{k,h}^{m+1} \boldsymbol{\psi}_i, \boldsymbol{\psi}_j \right) + \frac{1}{2} \nu_k \left( \alpha_{k,h}^{m+1} D(\boldsymbol{\psi}_i), D(\boldsymbol{\psi}_j) \right) + \rho_g^{-1} C_D \left( \alpha_{g,h}^{m+1} |\zeta_g - \zeta_l| \boldsymbol{\psi}_i, \boldsymbol{\psi}_j \right)$$

$$A_{mix_{ij}} = -\rho_g^{-1} C_D \left( \alpha_{g,h}^{m+1} |\zeta_g - \zeta_l| \boldsymbol{\psi}_i, \boldsymbol{\psi}_j \right)$$

$$B_{n,i} = \begin{pmatrix} -\left( \phi_n, \nabla \cdot (\rho_g^{-1} \alpha_{g,h}^{m+1} \boldsymbol{\psi}_i) \right) \\ -\left( \phi_n, \nabla \cdot (\rho_l^{-1} \alpha_{l,h}^{m+1} \boldsymbol{\psi}_i) \right) \end{pmatrix}.$$

**Proposition 1.** *The linear system* (56) *is uniquely solvable.*

**Proof.** According to the LBB theorem [46] the saddle point problem (52) is well posed when for $p \in P_h^{m+1}/\mathbb{R}$, there exists $v \in V_{0h}^{m+1}$ such that

$$\frac{(p, \nabla \cdot v)}{\|v\|_{H^1}} \geq c\|p\|_{L^2/\mathbb{R}} \text{ for some } c > 0. \tag{58}$$

Therefore $\boldsymbol{\Phi}$ has full rank and is non singular. $\square$

*4.6. Iterative Process*

At each time step, (47), is solved first, then (48) and (49) is solved iteratively by using a semi-linearization of the nonlinear boundary terms. Then (50) and (51) is solved iteratively by a semi-linearization of the nonlinear terms; each block involves the solution of a well posed symmetric linear system. Finally $S^m$ is updated by (23). Algorithm 2 summarizes the procedure.

Note that the computational domain is $\Omega^m = \{(x,y), 0 < x < L, 0 < y < y^m(x)\}$.

---

**Algorithm 2:** Algorithm for solving the full system of equations.

**Data:** $\alpha_{g,h}^m$, $\alpha_{l,h}^m$, $\boldsymbol{u}_{g,h}^m$, $\boldsymbol{u}_{l,h}^m$, $p_h^m$, $c_{s,h}^m$, $c_{k,h}^m$, $c_{g,h}^m$, $E_{mix,h}^m$, and $y^m$

Set initial data $\alpha_{g,h}^0$, $\alpha_{l,h}^0$, $\boldsymbol{u}_{g,h}^0$, $\boldsymbol{u}_{l,h}^0$, $c_{s,h}^0$, $c_{k,h}^0$, $c_{g,h}^0$, $E_{mix,h}^0$;

**for** *m* **do**

    Solve (47) to get $\alpha_{g,h}^{m+1}$, $\alpha_{l,h}^{m+1}$;

    Initial guess: $E_{mix,h}^{m+1,0} = E_{mix,h}^m$, $C_h^{m+1,0} =$ solution to (48) when mixed potential is $E_{mix,h}^{m+1,0}$;

    **while** $\|C_h^{m+1,k+1} - C_h^{m+1,k}\| \geq$ *tolerance* **do**

        Initial guess: $E_{mix,h}^{m+1,k+1,0} = E_{mix,h}^{m+1,k}$;

        **while** $\|E_{mix,h}^{m+1,k+1,l+1} - E_{mix,h}^{m+1,k+1,l}\|_{L^2(S^m)} \geq$ *tolerance* **do**

            Solve (48) to get $c_{s,h}^{m+1,k+1,l+1}$, $c_{k,h}^{m+1,k+1,l+1}$, $c_{g,h}^{m+1,k+1,l+1}$;

            Solve (49) to get $E_{mix,h}^{m+1,k+1,l*}$;

            $E_{mix,h}^{m+1,k+1,l+1} = \xi E_{mix,h}^{m+1,k+1,l*} + (1 - \xi) E_{mix,h}^{m+1,k+1,l}$, $0 < \xi < 1$;

        **end**

    **end**

    Solve (50) and (51) to get, $\boldsymbol{u}_{g,h}^{m+1}$, $\boldsymbol{u}_{l,h}^{m+1}$, $p_h^{m+1}$ (Using Algorithm 1);

    Update the mesh by $y^{m+1} = y^m + \delta t u_{2_{g,h}}^{m+1}$. A Gauss-Seidel smoother is applied if oscillations occur: $y^{m+1}(x)$ is averaged with its neighbors, $y^{m+1}(x - dx)$ and $y^{m+1}(x + dx)$.

**end**

---

## 5. Numerical Simulation

### 5.1. One-Dimensional Electroless Nickel Plating Problem

Here we reproduce, with a two-dimensional computation, the one-dimensional study by Kim and Sohn [8]. In their work, the electroless nickel plating process on a rotating disk with constant angular velocity is considered. In this situation, the velocity field near the surface of the rotating disk can be approximated by a uniformly distributed flow towards the plating surface. In addition, the thickness of the diffusion layer is assumed uniform on the surface. Consequently, for the modeling, the domain becomes one-dimensional. Given that the gas generation is not considered and only the steady state is computed in [8], a single-phase recovery $r_l = 1$, $c_g = 0$ is applied. Finally, four partial reactions in the electroless nickel plating process are considered:

$$H_2PO_2^- + H_2O = H_2PO_3^- + 2H^+ + 2e^- \quad \text{(anodic)} \tag{59}$$

$$H_2PO_2^- + 2H^+ + e^- = P + 2H_2O \quad \text{(cathodic)} \tag{60}$$

$$Ni^{2+} + 2e^- = Ni \quad \text{(cathodic)} \tag{61}$$

$$2H^+ + 2e^- = H_2 \quad \text{(cathodic)} \tag{62}$$

All chemical species are labeled as follows: $c_1$ is the concentration of the anodic hypophosphite ($H_2PO_2^-$), $c_2$ the concentration of the cathodic hypophosphite, $c_3$ the concentration of the nickel ion ($Ni^{2+}$), and $c_4$ the concentration of the hydrogen ion ($H^+$). Now the two-dimensional analog can be formulated: Let $\Omega = (0, \delta_3) \times (0, \epsilon)$, where $\delta_3$ is the thickness of the diffusion layer for nickel and $\epsilon << \delta_3$ is a small positive number. The thickness of the diffusion layer for species $j$ is given in [32]:

$$\delta_j = 1.61 D_j^{1/3} \omega^{-1/2} \nu^{1/6}. \tag{63}$$

The governing equation for the concentration profile is given by

$$\partial_t c_j + \boldsymbol{u} \cdot \nabla c_j - D_j \Delta c_j = 0 \quad \text{in } \Omega, \tag{64}$$

subject to the boundary conditions

$$c_j = c_{0j} \text{ at } x = \delta_3, \quad -D_j \frac{\partial c_j}{\partial n} = 0 \text{ at } y = 0, \epsilon,$$

$$-D_j \frac{\partial c_j}{\partial n} = \frac{|i_1(E_{mix})|}{z_1 F} \left( \frac{(1-r)c_1}{c_{01}} \right)^{\gamma_1} + \frac{|i_2(E_{mix})|}{z_2 F} \left( \frac{rc_2}{c_{02}} \right)^{\gamma_2} \quad j = 1, 2, \tag{65}$$

$$-D_j \frac{\partial c_j}{\partial n} = \frac{|i_j(E_{mix})|}{z_j F} \left( \frac{c_1}{c_{01}} \right)^{\gamma_j} \quad j = 3, 4 \text{ at } x = 0.$$

with the electron balance constraint

$$\sum_{j=1}^{4} \frac{i_j(E_{mix})}{z_j F} \left( \frac{c_j}{c_{0j}} \right)^{\gamma_j} = 0. \tag{66}$$

The velocity field can be expressed as in [32]:

$$\boldsymbol{u} = (-a x^2 \omega^{3/2} \nu^{-1/2}, 0)^T \tag{67}$$

where $a = 0.51023$ is an experimental constant, $r = 0.995$ is the ratio between the hypophosphite anodic part and the cathodic part on the reacting surface. The equilibrium potential $E_{0j}$ for species $j$ can be approximated by the Nernst equation, with $pH = \log(c_{04})$:

$$E_{01} = -0.878 + \frac{0.25R\theta}{F}\log\left(10^{4.5}c_{04}\right), \quad E_{02} = -0.806 + \frac{0.3R\theta}{F}\log\left(10^{4.5}c_{04}\right),$$

$$E_{03} = -0.147, \quad E_{04} = -0.101 + \frac{R\theta}{F}\log\left(10^{4.5}c_{04}\right). \tag{68}$$

By simulating system (64), (65), and (66), with the physical constants given in Tables 1 and 2, until a steady state is reached, the numerical tests show that the present model agrees well with the previous 1D studies of Kim and Sohn [8]: see Figures 6 and 7.

**Table 1.** Physical parameters used in [8], valid for pH $= 4.5$ and the concentration of $H_2PO_2^- = 0.3$ M.

|  | $H^+$ | $Ni^{2+}$ | $H_2PO_2^-$ (Cathodic) | $H_2PO_2^-$ (Anodic) |
|---|---|---|---|---|
| $i_0$ (A/cm$^2$) | $1.5 \times 10^{-4}$ [a] | $1.5 \times 10^{-7}$ [b] | $6.0 \times 10^{-4}$ | $8.9 \times 10^{-3}$ |
| $D$ (cm$^2$/s) | $4.5 \times 10^{-5}$ [a] | $0.5 \times 10^{-6}$ [c] | $1.7 \times 10^{-5}$ | $1.7 \times 10^{-5}$ |
| $\alpha$ | 0.79 [a] | 0.79 [c] | 0.2 | 0.9 |
| $\beta$ | 0.21 [a] | 0.21 [c] | 0.8 | 0.1 |
| $z$ | 1 | 2 | 1 | 4 |
| $\gamma$ | 1.0 [a] | 1.0 [c] | 0.3 | 1.0 |
| $E_0$ (V) [d] | $-0.101$ | $-0.147$ | $-0.806$ | $-0.878$ |
| $c_0$ (M) | $3.162 \times 10^{-5}$ [e] | 0.1 | 0.3 | 0.3 |
| subscript $j$ | 4 | 3 | 2 | 1 |

[a] Estimated from [49]. [b] Assumed in this study. [c] Taken from [50]. [d] Calculated based on [51]. All values except $E_{03}$ ($Ni^{2+}$) depend on pH (see (68)). [e] If pH $= x$, then $c_{04} = 10^{-x}$ (M). $i_0$: Exchange current density, $D$: Diffusion coefficient. $\alpha$: Anodic transfer coefficient, $\beta$: Cathodic transfer coefficient. $z$: Number of electron transport, $\gamma$: Reaction order, $E_0$: Equilibrium potential (90 °C) $c_0$: inlet and initial concentration.

**Table 2.** Conditions assumed in [8] for performing our simulations.

| **Experimental Conditions** | |
|---|---|
| Angular velocity $\omega$ | 400 rpm |
| Kinematic viscosity $\nu$ | $1.2 \times 10^{-2}$ cm$^2$/s |
| Temperature $\theta$ | 90 °C |
| **Composition of Electrolytes** | |
| NiSO$_4$ (nickel sulfate) | 0.1 M |
| NaH$_2$PO$_2$ (sodium hypophosphite) | 0.3 M |
| pH | 4.0–5.3 |

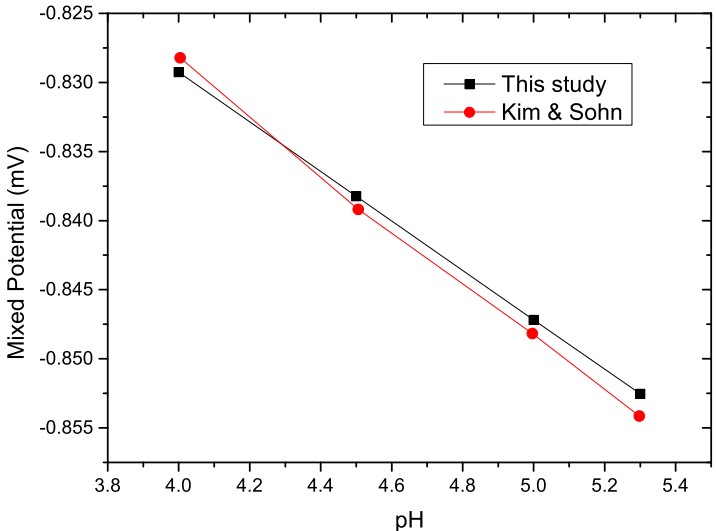

**Figure 6.** In red, the mixed potential $E_{mix}$ computed by the one dimensional system (65) versus pH $= \log c_{04}$. In black, the same but computed with the full two dimensional system.

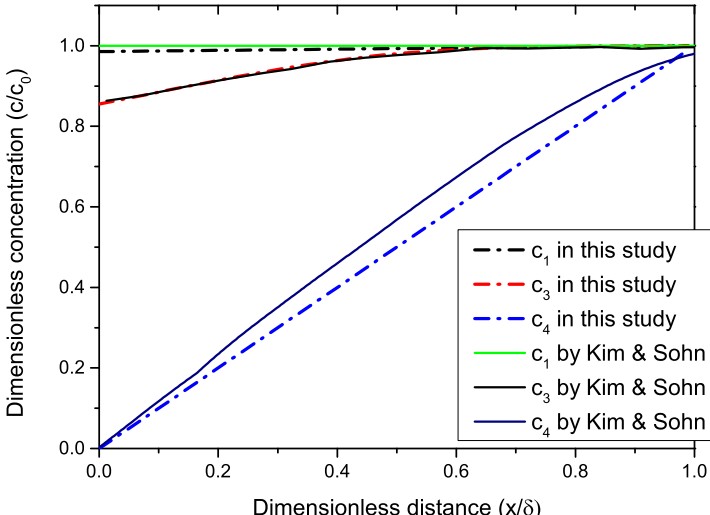

**Figure 7.** Concentration profiles of three chemical species versus $x$ computed by the one-dimensional system (65) and compared with the results of the full two-dimensional system.

Regarding (59)–(62), atomic nickel and phosphorus are deposited on the surface during the electroless process. The deposition thickness can be estimated in terms of the current densities:

$$\left( \frac{i_2(E_{mix})(c_2|_{x=0}/c_{02})V_P}{z_2 F} + \frac{i_3(E_{mix})(c_3|_{x=0}/c_{03})V_{Ni}}{z_3 F} \right) t, \tag{69}$$

where $V_P$, $V_{Ni}$ are molar volumes of phosphorus and nickel, respectively, and $t$ is the deposition time.

### 5.2. Two Species in a Gas–Liquid Two-Phase Flow

Let the initial domain $\Omega$ be a rectangle of size $0.01 \times 0.001$ (in meters). We consider complexed (by tartrate, denoted by $L$) copper ions, formaldehyde, and hydrogen dissolved in water, which are denoted by the subscriptions $s, k, g$, respectively, for the chemical species transport equations.

The chemical reaction can be expressed as the following two partial reactions:

$$Cu(OH)_2L_2^{-4} = Cu + 2OH^- + 2L^{-2} \tag{70}$$

$$2HCHO + 4OH^- = 2HCOO^- + H_2 + 2H_2O + 2e^- \tag{71}$$

Given the above equations, we also use the subscriptions $s$ and $k$ to represent the quantities corresponding to (70) and (71), respectively.

The values of the physical constants are listed in Table 3.

For convenience, the following scalings are applied:

$$
\begin{aligned}
& L \to \frac{L}{d_0} \ \ (L \text{ is any length}), \ \ \rho_l \to \frac{\rho_l}{\rho_0}, \ \ \rho_g \to \frac{\rho_g}{\rho_0}, \\
& \mu_l \to \frac{\mu_l}{\rho_0 d_0 u_0}, \ \ \mu_g \to \frac{\mu_g}{\rho_0 d_0 u_0}, \ \ c_g \to \frac{c_g}{c_{g0}}, \ \ c_k \to \frac{c_k}{c_{k0}}, \ \ c_s \to \frac{c_s}{c_{s0}}, \ \ K \to \frac{d_0 c_{g0}}{\rho_0 u_0} K, \\
& L_g \to \frac{L_g}{u_0 c_{g0} z_s F}, \ \ L_s \to \frac{L_s}{u_0 c_{s0} z_s F}, \ \ L_k \to \frac{L_k}{u_0 c_{k0} z_k F}, \ \ D_g \to \frac{D_g}{u_0 d_0}, \ \ D_s \to \frac{D_s}{u_0 d_0}, \ \ D_k \to \frac{D_k}{u_0 d_k}, \\
& i_g \to \frac{i_g}{u_0 c_{g0} z_s F}, \ \ i_s \to \frac{i_s}{u_0 c_{s0} z_s F}, \ \ i_k \to \frac{i_k}{u_0 c_{k0} z_k F}.
\end{aligned} \tag{72}
$$

**Table 3.** Parameters used in Section 5.2.

| Physical Quantity | Value | Physical Quantiy | Value |
|---|---|---|---|
| $\rho_l$ (kg/m$^3$) | 995.65 | $\rho_g$ (kg/m$^3$) | 1.161 |
| $\rho_0$ (kg/m$^3$) | 1.161 | $\mu_g$ (kg/m·s) | $1.86 \times 10^{-5}$ |
| $\mu_l$ (kg/m·s) | $7.977 \times 10^{-4}$ | $d_0$ (m) | 0.001 |
| $u_0$ (m/s) | 0.001 | $c_{sat}$ (mol/m$^3$) | 0 |
| $i_g$ (A/m$^2$) | $1.0 \times 10^{-2}$ | $i_s$ (A/m$^2$) | $1.0 \times 10^{-2}$ |
| $i_k$ (A/m$^2$) | 10 | $R$ (J/K·mol) | 8.314 |
| $K$ (kg/mol·s) | $3.87 \times 10^{-4}$ | $M_g$ (mol/kg) | 500 |
| $c_{g,0}$ (mol/m$^3$) | 1 | $c_{s,0}$ (mol/m$^3$) | 39.34 |
| $c_{k,0}$ (mol/m$^3$) | 77.58 | $D_g$ (m$^2$/s) | $2 \times 10^{-8}$ |
| $D_s$ (m$^2$/s) | $7 \times 10^{-10}$ | $D_k$ (m$^2$/s) | $1.2 \times 10^{-9}$ |
| $z_s$ (1) | 2 | $z_k$ (1) | 4 |
| $\alpha_s$ (1) | 0.67 | $\alpha_k$ (1) | 0.37 |
| $\beta_s$ (1) | 0.33 | $\beta_k$ (1) | 0.63 |
| $\theta$ (K) | 363.15 | $E_s$ (V) | $-0.266$ |
| $E_k$ (V) | 1.5 | $C_D$ (1) | 242220 |
| $\alpha$ (1) | 0.0005 | | |

The initial conditions are set to: constant phase ratio and Poiseuille flow:

$$r_g^0 = \epsilon, \ \ r_l^0 = 1 - \epsilon, \ \ \boldsymbol{u}_g^0 = \boldsymbol{u}_l^0 = (0.69y(1-y), 0)^T, \tag{73}$$

with $\epsilon = 0.0001$. Additionally, let $C^0 = (c_s^0, c_k^0, c_g^0)^T$ satisfies

$$-\nabla \cdot (r_l^0 \boldsymbol{D} \nabla C^0) = 0, \ \ C^0|_{\Gamma_{in}} = (1,1,0)^T, \ \ \frac{\partial C^0}{\partial n}|_{\Gamma_{out} \cup \Gamma_{wall}} = 0 \tag{74}$$

plus the first equation in (76) subject to (21). The inflow values are

$$\boldsymbol{u}_g^{m+1}|_{\Gamma_{in}} = \boldsymbol{u}_l^{m+1}|_{\Gamma_{in}} = (y(1-y), 0)^T, \ \ c_g|_{\Gamma_{in}} = 0, \ \ c_s|_{\Gamma_{in}} = c_k|_{\Gamma_{in}} = 1, \ \ r_l|_{\Gamma_{in}} = 1 - \epsilon. \tag{75}$$

The boundary conditions on $S(t^{m+1})$ are

$$-D_p \frac{\partial c_p^{m+1}}{\partial n} = \chi r_l^{m+1} |i_p^{m+1}| c_p^{m+1}, \ \ p = s, g, k, \ \ \boldsymbol{u}_g^{m+1} = \boldsymbol{u}_l^{m+1} = \alpha \chi r_l^{m+1} |i_s^{m+1}| c_s^{m+1}, \tag{76}$$

where $\alpha = 0.0005$ and

$$\chi(x,y) = \begin{cases} x - \frac{3}{4} + \frac{1}{4} \sin\left(2\pi(x - \frac{1}{4})\right), & 1 \leq x < 1.5, \\ 1, & 1.5 \leq x < 5.5, \\ \frac{17}{4} - x - \frac{1}{4} \sin\left(2\pi(x - \frac{19}{4})\right), & 5.5 \leq x < 6, \\ 0, & 0 \leq x < 1 \text{ or } 6 \leq x \leq 10. \end{cases} \tag{77}$$

Boundary conditions on $\Gamma_{out}$ and $\Gamma_{wall}$ are as in Section 2.4. See also Figure 4.

**Remark 3.** *We note that $\alpha = 0.0005$ is much larger than the experimental values; the numerical simulations produce $u_{g_2}$ (and $u_{l_2}$) of magnitude in the order $O(10^{-4})$. On the other hand, the deposition rate in a typical experiment is of order 1 μm per hours [52], which is not larger than $O(10^{-6})$. Yet the numerical test is conducted to validate the numerical method when the evolution of the domain is larger than real-life values.*

### 5.2.1. Convergence

First, we conduct the convergence test for different time step with a fixed mesh. To obtain a "reference solution", the system (47)–(51) is solved with a $50 \times 10$ uniform mesh and a small time step $\delta t = 0.01$ and $T = 10$. The convergence with respect to $\delta t$ is studied without changing the mesh; results are given in Table 4 and the rate of convergence for each variable is presented in Figure 8. Numerical tests for solving two-phase flow problem and volume fraction problem present a linear decay of $L^2$ error with respect to the time step. However, the convergence for solving the concentration profiles does not reach the expectation due to extremely low diffusion coefficients and large current densities.

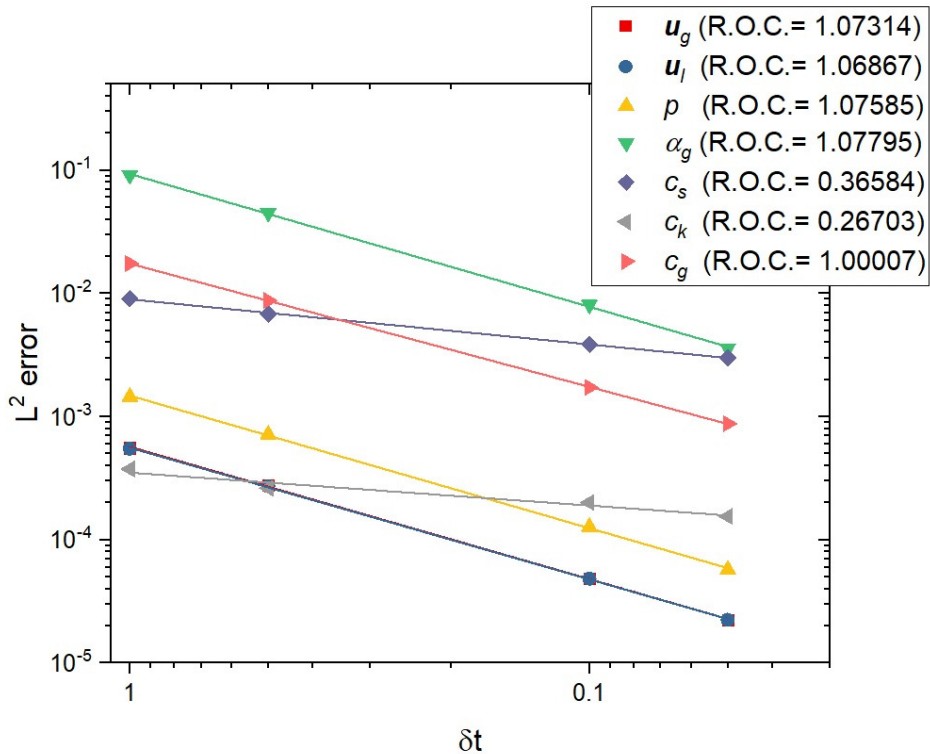

**Figure 8.** Convergence with respect to $\delta t$ for the case described in Section 5.2; log–log plot of the error for each unknown (note that the curves for $u_g$ and $u_l$ overlap). R.O.C. means "Rate Of Convergence". The reference solution is a computation with a very small time step.

**Table 4.** $L^2$ error with respect to the reference solution provided with time step $\delta t = 0.01$, $50 \times 10$ uniform mesh, and $\alpha = 0.0005$ for numerical simulation in Section 5.2 at $T = 10$.

| $\delta t$ | $u_g$ | $u_l$ | $p$ | $\alpha_g$ | $c_s$ | $c_k$ | $c_g$ |
|---|---|---|---|---|---|---|---|
| 1 | $5.56 \times 10^{-4}$ | $5.49 \times 10^{-4}$ | $1.45 \times 10^{-3}$ | $9.11 \times 10^{-2}$ | $9.12 \times 10^{-3}$ | $3.77 \times 10^{-4}$ | $1.74 \times 10^{-2}$ |
| 0.5 | $2.77 \times 10^{-4}$ | $2.73 \times 10^{-4}$ | $7.17 \times 10^{-4}$ | $4.50 \times 10^{-2}$ | $6.80 \times 10^{-3}$ | $2.64 \times 10^{-4}$ | $8.74 \times 10^{-3}$ |
| 0.1 | $4.85 \times 10^{-5}$ | $4.84 \times 10^{-5}$ | $1.28 \times 10^{-4}$ | $8.07 \times 10^{-3}$ | $3.87 \times 10^{-3}$ | $2.01 \times 10^{-4}$ | $1.72 \times 10^{-3}$ |
| 0.05 | $2.25 \times 10^{-5}$ | $2.24 \times 10^{-5}$ | $5.75 \times 10^{-5}$ | $3.58 \times 10^{-3}$ | $3.00 \times 10^{-3}$ | $1.55 \times 10^{-4}$ | $8.77 \times 10^{-4}$ |

Second, we conduct the convergence tests for different time steps and mesh pairs; the time step is always proportional to the mesh size. The reference solution is obtained with $200 \times 20$ uniform mesh and $\delta t = 0.05$ at $T = 10$. Figure 9 and Table 5 present a linear decay of $L^2$ error with respect to the time step for each variable.

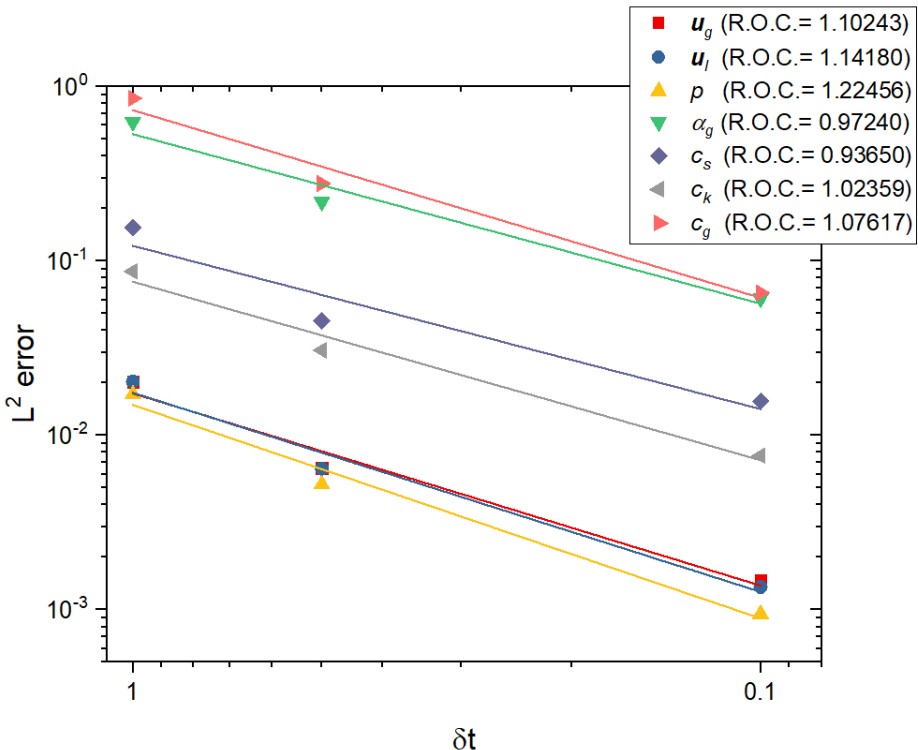

**Figure 9.** Convergence with respect to $\delta t$ and mesh size for the case described in Section 5.2 (see Table 5 for the time step and mesh size pair): log–log plot of the error for each unknown (note that the curves for $u_l$ and $u_l$ overlap).

**Table 5.** $L^2$ error with respect to the reference solution provided with time step $\delta t = 0.05$, $200 \times 20$ uniform mesh, and $\alpha = 0.0005$ for numerical simulation in Section 5.2 at $T = 10$.

| $(\delta t, \text{Mesh})$ | $u_g$ | $u_l$ | $p$ | $\alpha_g$ | $c_s$ | $c_k$ | $c_g$ |
|---|---|---|---|---|---|---|---|
| $(1, 25 \times 3)$ | $2.03 \times 10^{-2}$ | $2.04 \times 10^{-2}$ | $1.72 \times 10^{-2}$ | $6.24 \times 10^{-1}$ | $1.55 \times 10^{-1}$ | $8.73 \times 10^{-2}$ | $8.57 \times 10^{-1}$ |
| $(0.5, 50 \times 5)$ | $6.50 \times 10^{-3}$ | $6.45 \times 10^{-3}$ | $5.24 \times 10^{-3}$ | $2.18 \times 10^{-1}$ | $4.54 \times 10^{-2}$ | $3.08 \times 10^{-2}$ | $2.78 \times 10^{-1}$ |
| $(0.1, 100 \times 10)$ | $1.47 \times 10^{-3}$ | $1.35 \times 10^{-3}$ | $9.48 \times 10^{-4}$ | $6.09 \times 10^{-2}$ | $1.57 \times 10^{-2}$ | $7.65 \times 10^{-3}$ | $6.58 \times 10^{-2}$ |

Third, a convergence test similar to the second test above is performed with different time steps and mesh pairs, but the errors are computed at $T = 120$. The reference solution is obtained with $200 \times 20$ uniform mesh and $\delta t = 0.3$. Figure 10 and Table 6 show a nearly linear decay of $L^2$ error with respect to the time step for each variable. In this test, $r_g$ and $c_k$ vary significantly as the grid is refined. The intensity maps of $r_g$ are given in Figure 11. The intensity maps of all variables present no significant differences with the reference solutions when $\delta t \leq 0.6$.

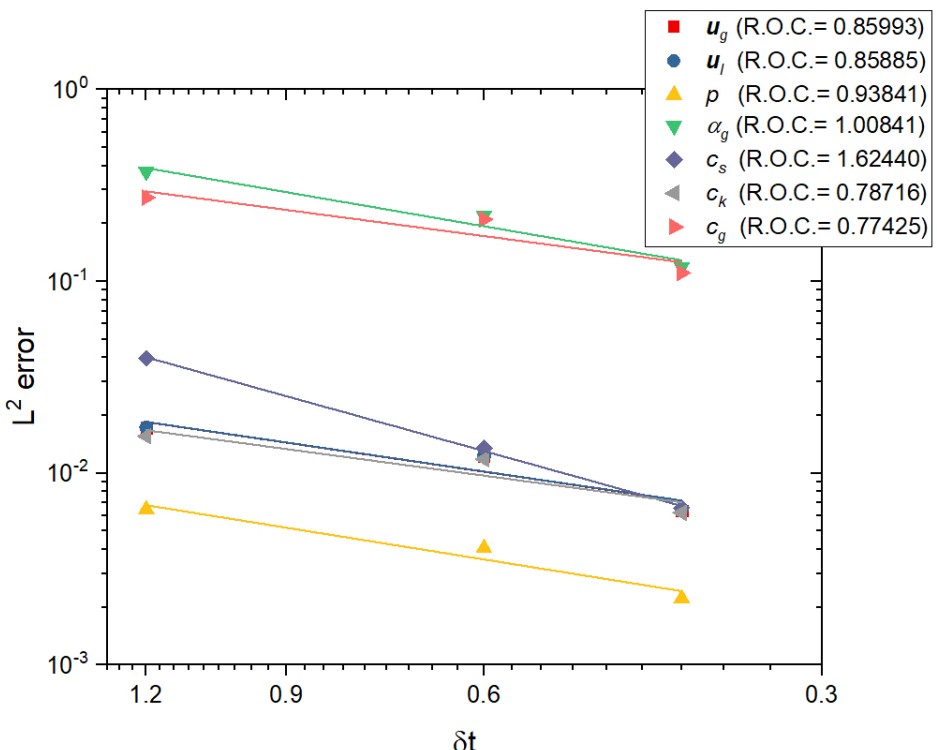

**Figure 10.** Convergence with respect to $\delta t$ and mesh size for the case described in Section 5.2 (see Table 5 for the time step and mesh size pair): log–log plot of the error for each unknown (note that the curves for $u_g$ and $c_k$ overlap and the curve of $u_l$ is closed to them).

**Table 6.** $L^2$ error with respect to the reference solution provided with time step $\delta t = 0.3$, $200 \times 20$ uniform mesh, and $\alpha = 0.0005$ for numerical simulation in Section 5.2 at $T = 120$.

| $(\delta t, \textbf{Mesh})$ | $u_g$ | $u_l$ | $p$ | $\alpha_g$ | $c_s$ | $c_k$ | $c_g$ |
|---|---|---|---|---|---|---|---|
| $(1.2, 50 \times 5)$ | $1.73 \times 10^{-2}$ | $1.73 \times 10^{-2}$ | $6.48 \times 10^{-3}$ | $3.71 \times 10^{-1}$ | $3.97 \times 10^{-2}$ | $1.56 \times 10^{-2}$ | $2.73 \times 10^{-1}$ |
| $(0.6, 100 \times 10)$ | $1.22 \times 10^{-2}$ | $1.23 \times 10^{-2}$ | $4.07 \times 10^{-3}$ | $2.20 \times 10^{-1}$ | $1.35 \times 10^{-2}$ | $1.18 \times 10^{-2}$ | $2.11 \times 10^{-1}$ |
| $(0.4, 150 \times 15)$ | $6.40 \times 10^{-3}$ | $6.41 \times 10^{-3}$ | $2.23 \times 10^{-3}$ | $1.19 \times 10^{-1}$ | $6.60 \times 10^{-3}$ | $6.23 \times 10^{-3}$ | $1.11 \times 10^{-1}$ |

### 5.2.2. Robustness for Large Time Steps

With a large time step $\delta t = 1$ and $100 \times 10$ uniform mesh, the product of the maximal liquid fluid speed with the time step is around 1.5 times of the mesh size, which is optimal for the Galerkin-characteristic method. Solutions are displayed in Figures 12–16.

### 5.2.3. CPU Time

With $\delta t = 1$ and $100 \times 10$ uniform mesh, it took 5832 s to reach the final time $T = 180$ with an Intel Core i7-8750H @ 2.20 GHz. During the computation, it took 0.086% of the total CPU to solve the volume fraction problem, 7.66% to solve the chemical species transport problem, and 91.93% to solve the two-velocities/pressure flow problem.

The computer program is written using the `FreeFEM++` toolkit [35].

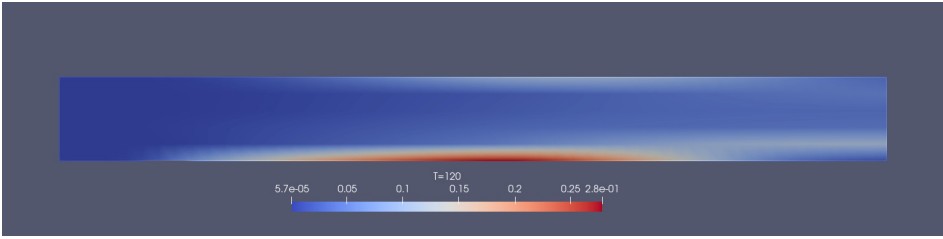

(**a**) Intensity map of $r_g$ at $t = 120$ with $\delta t = 1.2$ and $50 \times 5$ uniform mesh.

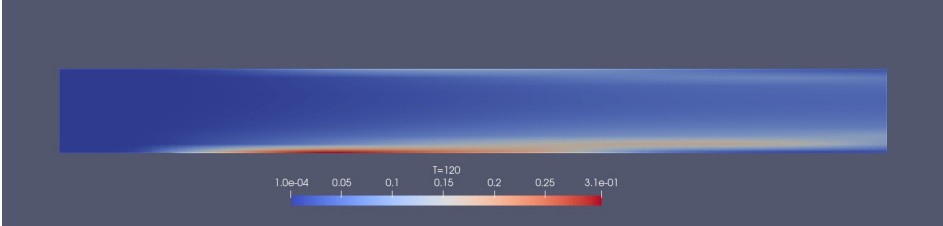

(**b**) Intensity map of $r_g$ at $t = 120$ with $\delta t = 0.6$ and $100 \times 10$ uniform mesh.

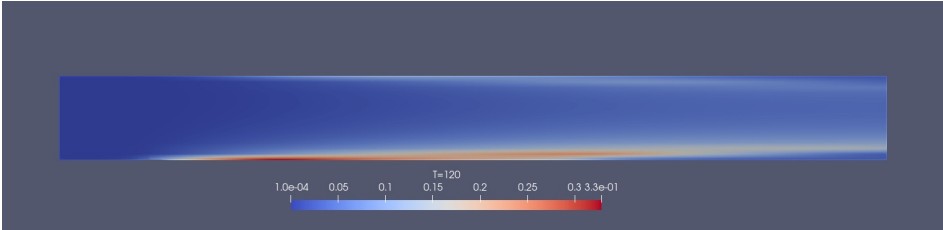

(**c**) Intensity map of $r_g$ at $t = 120$ with $\delta t = 0.4$ and $150 \times 15$ uniform mesh.

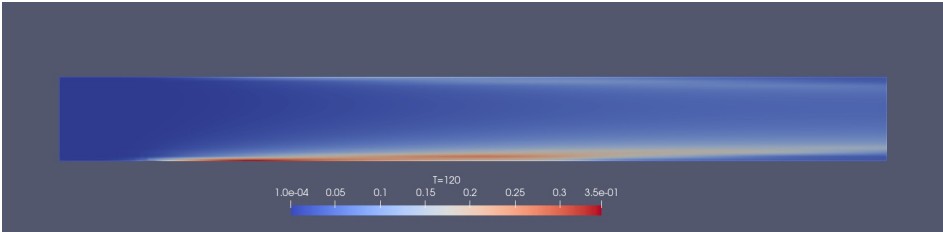

(**d**) Intensity map of $r_g$ at $t = 120$ with $\delta t = 0.3$ and $200 \times 20$ uniform mesh.

**Figure 11.** For Section 5.2: The intensity maps of $r_g$ for different time step and mesh size pairs.

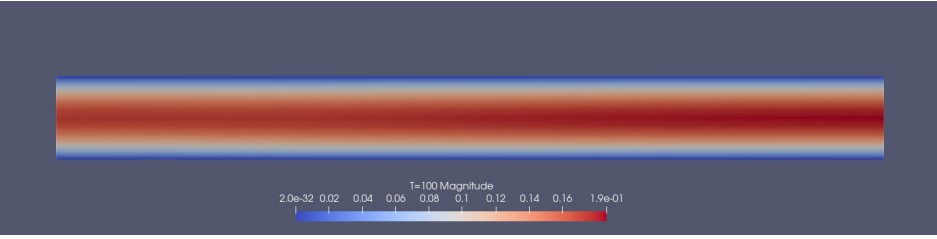

(**a**) Intensity map of $u_g$ at $t = 100$.

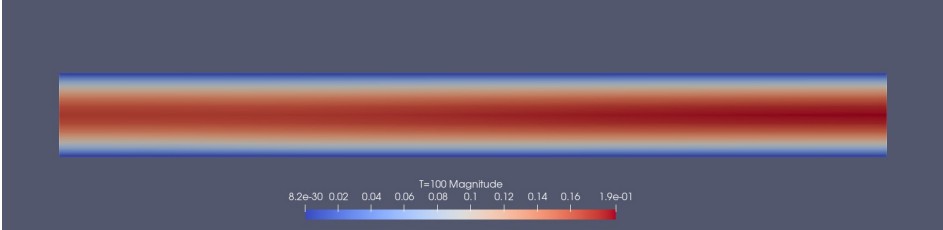

(**b**) Intensity map of $u_l$ at $t = 100$.

**Figure 12.** For Section 5.2: The velocity magnitudes of $u_g$ and $u_l$.

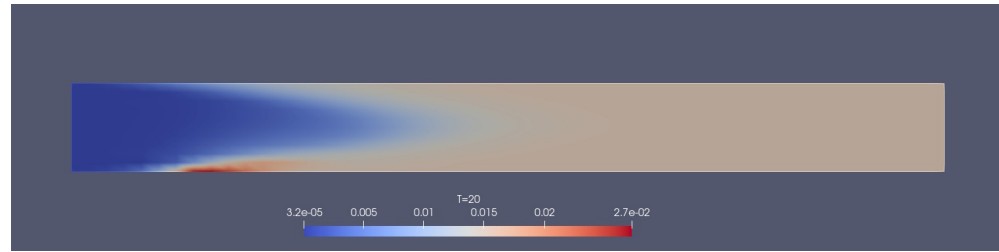

(**a**) Intensity map of $r_g$ at $t = 20$.

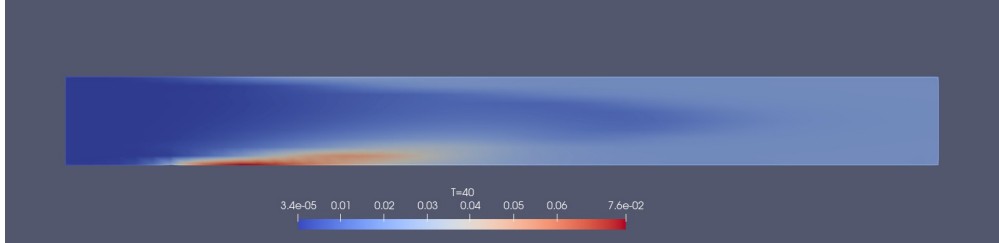

(**b**) Intensity map of $r_g$ at $t = 40$.

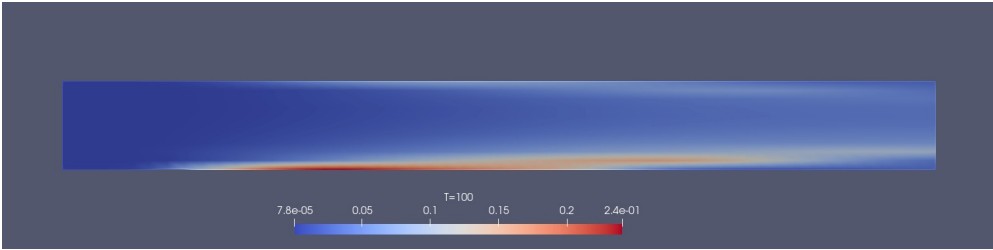

(**c**) Intensity map of $r_g$ at $t = 100$.

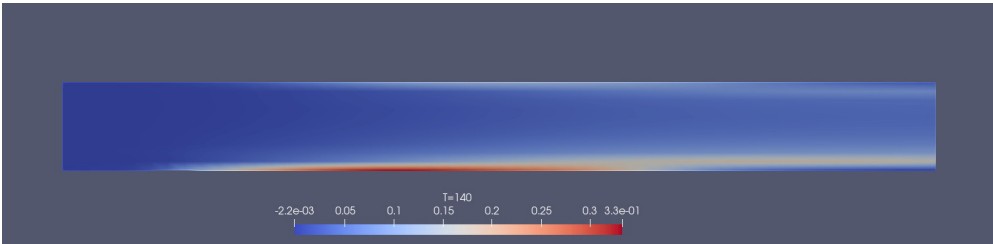

(**d**) Intensity map of $r_g$ at $t = 140$.

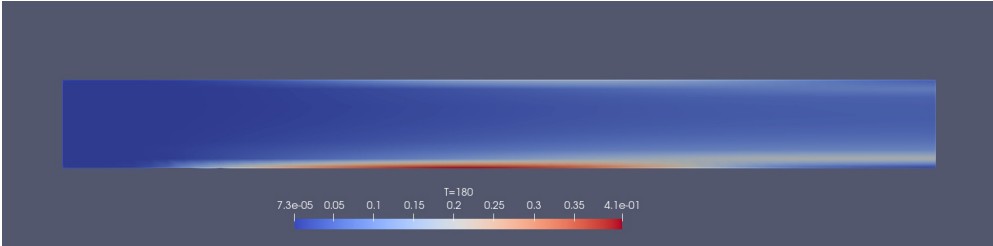

(**e**) Intensity map of $r_g$ at $t = 180$.

**Figure 13.** For Section 5.2: intensity maps of the volume fraction of the gas phase $r_g$ computed with $\delta t = 1$ and a $100 \times 10$ uniform mesh.

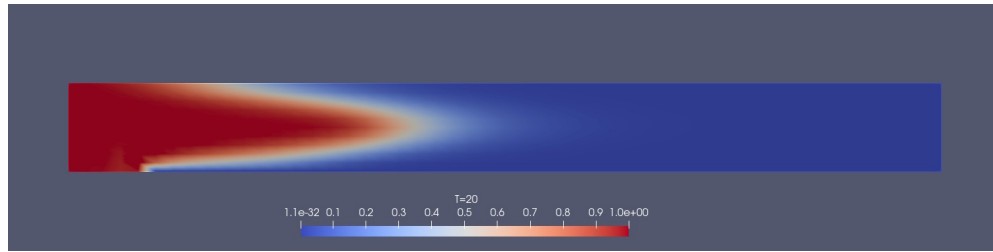

(**a**) Intensity map of $c_s$ at $t = 20$.

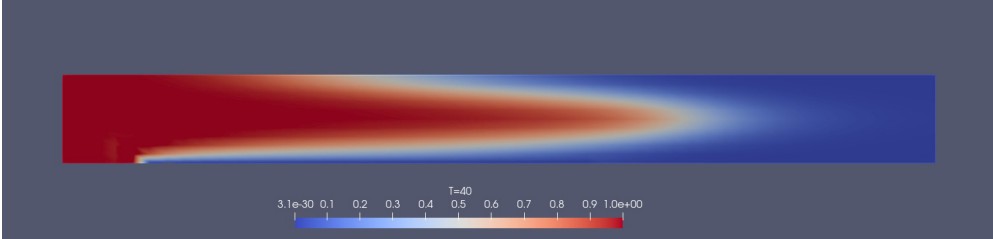

(**b**) Intensity map of $c_s$ at $t = 40$.

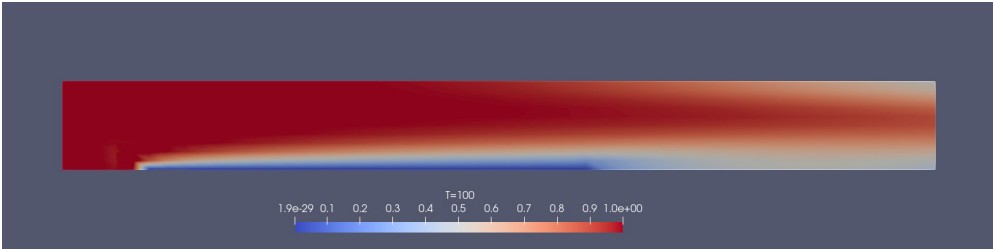

(**c**) Intensity map of $c_s$ at $t = 100$.

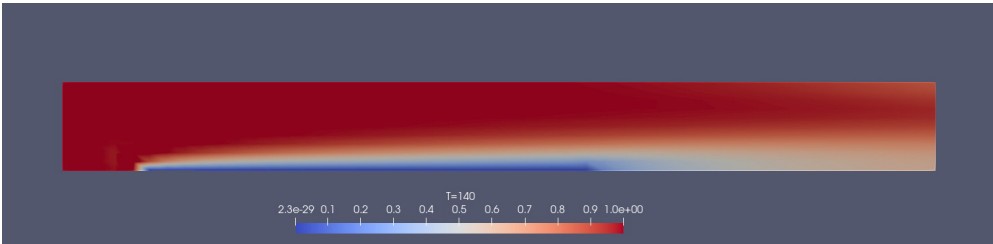

(**d**) Intensity map of $c_s$ at $t = 140$.

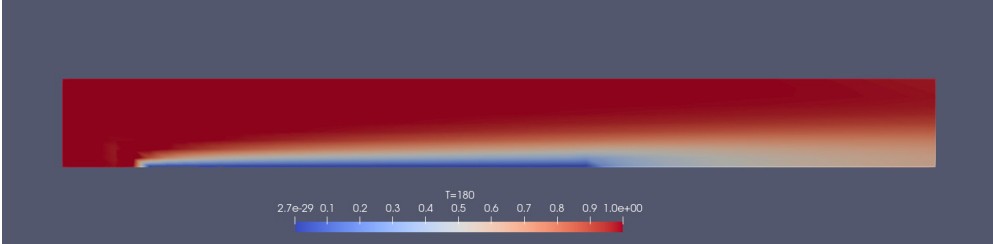

(**e**) Intensity map of $c_s$ at $t = 180$.

**Figure 14.** For Section 5.2: intensity maps of the concentration electrolyte ions $c_s$ computed with $\delta t = 1$ and a $100 \times 10$ uniform mesh. The blue zone in the plating region on the lower plate shows that the electrolyte is absorbed by the plating process.

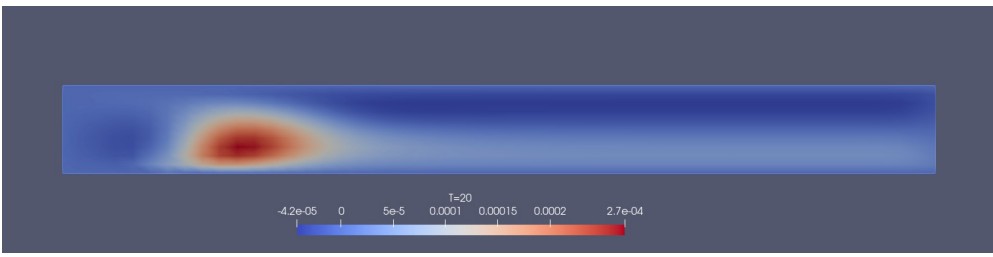

(**a**) Intensity map of $u_{g_2}$ at $t = 20$.

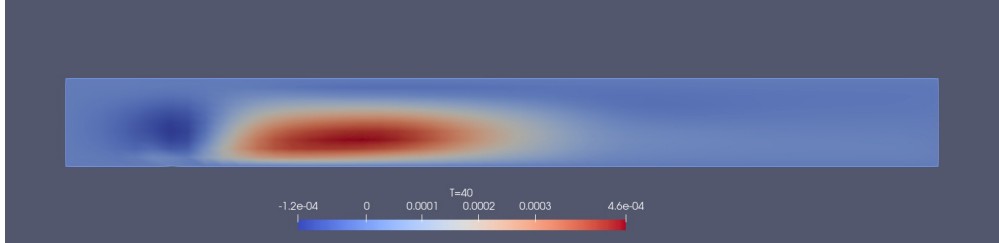

(**b**) Intensity map of $u_{g_2}$ at $t = 40$.

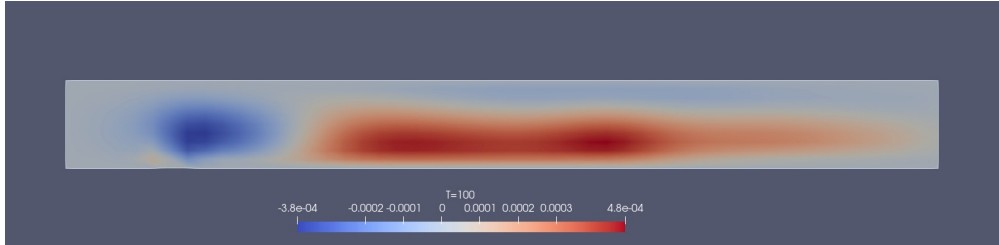

(**c**) Intensity map of $u_{g_2}$ at $t = 100$.

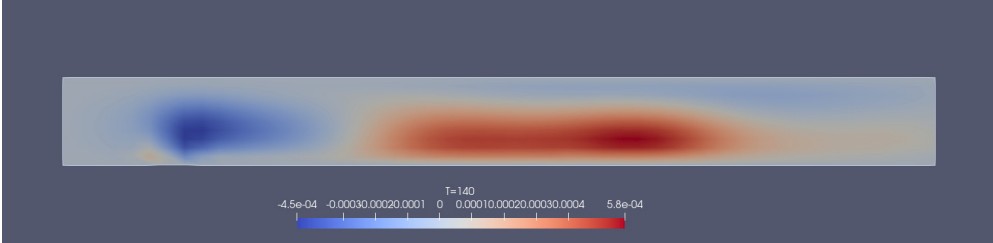

(**d**) Intensity map of $u_{g_2}$ at $t = 140$.

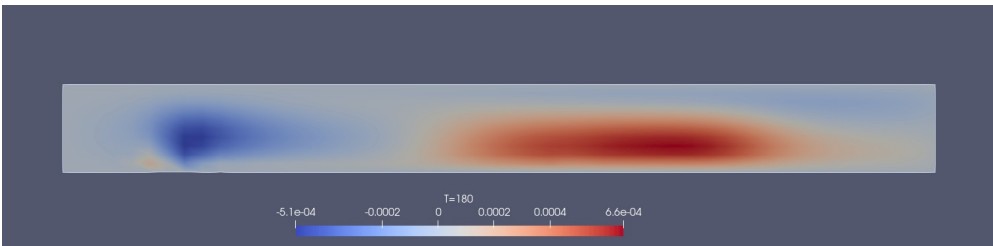

(**e**) Intensity map of $u_{g_2}$ at $t = 180$.

**Figure 15.** For Section 5.2: The vector fields $\boldsymbol{u}_g$ and $\boldsymbol{u}_l$ are very closed to Poiseuille flow. In this case, phase change and moving boundary contribute to the second component of $\boldsymbol{u}_g$ (and $\boldsymbol{u}_l$) together. The numerical test is conducted with $\delta t = 1$ and $100 \times 10$ uniform mesh. The intensity maps indicate the bubble rising in the red region. Indeed, there exists high gas volume fraction region near the top side (see Figure 13).

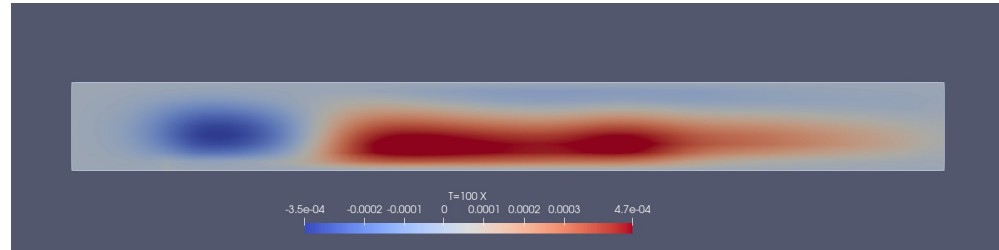

(**a**) Intensity map of $u_{l2}$ at $t = 100$.

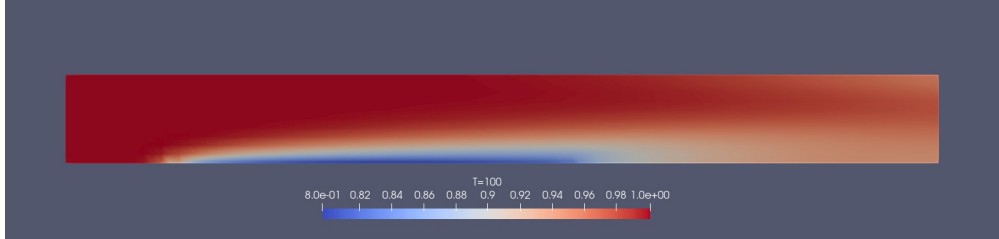

(**b**) Intensity map of $c_k$ at $t = 100$.

**Figure 16.** For Section 5.2: Intensity maps of $u_{l2}$ and $c_k$ at $t = 100$.

### 5.2.4. Results

In Figure 12a,b the velocity vector fields $\boldsymbol{u}_g$ and $\boldsymbol{u}_l$ are seen to be almost parabolic in $y$ (Poiseuille flow), but the phase change and moving boundary induce a non-zero asymmetric vertical component $u_{2g}$ (see Figure 15); both play important roles for the bubble distribution. Bubble density can be inferred by analyzing $c_g$ and $r_g$ (see Figure 3). The color maps of Figure 13 display a high gas volume fraction area near the top and bottom plates. Figure 14 shows how the steady state is established and how the electrolyte disappears in the plating region due to the plating. Figure 2 displays a high volume fraction of the gaseous phase near the reacting surface. The deposition-induced movement of $S$ is presented in Figure 17. Figure 3b shows that the region of highest bubble density is moving away from the inlet as the electroless plating proceeds.

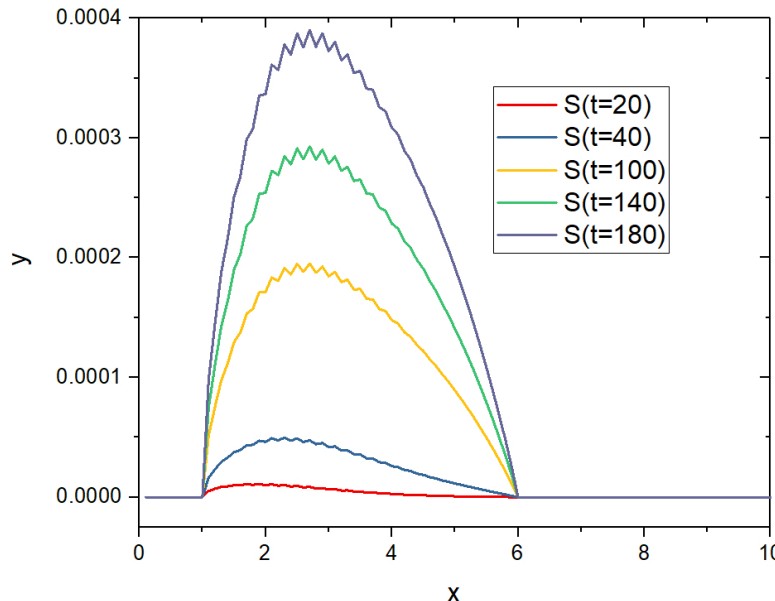

**Figure 17.** The thickness of the deposition is given by the motion of $S(t)$, plotted here at 5 instants of time, with respect to the $x$-axis (in mm). Notice that the motion $t \to S(t)$ is very small; the oscillations are blown-out of proportion by the scaling used in the graphic.

## 6. Comparison with Experimental Results

To validate the numerical method on a real-life problem, an experiment for reproducing the numerical study in Section 5.2 is conducted. Here, we shall show that the experimental result can be qualitatively fitted by the numerical simulation.

The experimental setting is described as the following: A micro-channel is enclosed by two sheet glasses of size 8 mm × 8 mm and another two of size 8 mm × 1 mm, which form a rectangular channel. The electrolyte goes in the channel from the left and exit on the right. One piece of the square sheet glasses is partially glued on a copper plate of size 8 mm × 4 mm, where the longer side of the copper plate coincides with an edge of the inlet (see Figure 1 for geometry setting). The inflow is set to be of average velocity 0.115 mm/s. At inlet, the copper ion concentration is $c_{s0} = 39.34$ mol/m$^3$ and the formaldehyde concentration is $c_{k0} = 77.5883$ mol/m$^3$. Here, the inlet concentrations $c_{g0}$ and $c_{k0}$ are the reference concentrations for copper ion and formaldehyde, respectively. We further define the reference concentration of the hydrogen gas to be $c_{g0} = 1$ mol/m$^3$. Other physical parameters are given in Table 3. Some parameters, for example, reference current densities $i_s, i_k,$ and $i_g$, may not be exactly same as what are given in Table 3. Nevertheless, they are acceptably closed to reality, or at least in the same order.

### 6.1. Experimental

To fabricate the test vehicle, a 4 inch glass wafer was first sputtered with 30 nm chromium and 200 nm copper which served as an adhesion layer and seed layer, respectively. The wafer was then diced into each 8 mm × 8 mm glass dies. To ensure a significant comparison between the regions being plated or not, each test die was half immersed in SPS ($Na_2S_2O_8$) solution and hydrochloric acid to remove the copper and chromium layer. The glass die turned out half transparent and half coated with copper where the electroless copper plating took place. Thereafter, a fully transparent glass, which was identical to the size of the test die, was face-to-face aligned and bonded via using a flip-chip die-bonder in order to obtain a clear observation view. Two tungsten wires which were 8 mm in length and 2 mm in diameter were glued by UV gel and placed on the periphery of the test die for the purpose of restricting the flow direction and defining the height between the dies (see Figure 18).

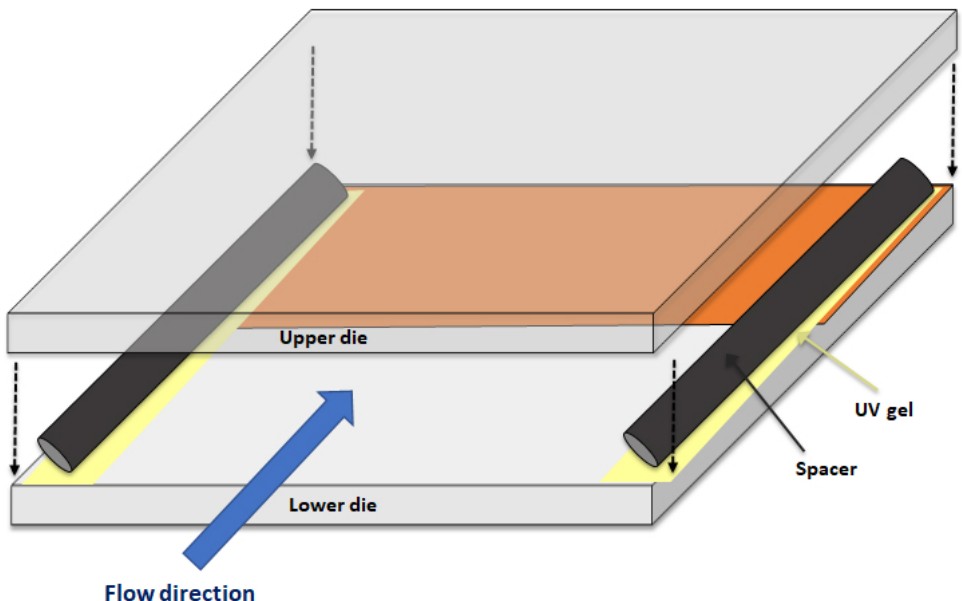

**Figure 18.** Test vehicle formation.

The test vehicle was then subjected to a micro-fluidic system composed of a PDMS mode containing a micro-fluidic channel and a bottom glass. Clips were used to seal the micro-fluidic system and prevent the leakage of electrolyte. A peristatic pump was used to control the flow and connect the micro-fluidic system with a silicone tube. Prior to the electroless plating, the test vehicle was immersed in 10% sulfuric acid to remove copper oxide. Finally, the electroless copper plating was conducted in a water tank controlled at 50 °C with in situ recording via stereomicroscope (charged coupled device digital camera CCD). The electrolyte PHE-1 Uyemura possessing the given reference concentrations $c_{s0}$ of (complexed) copper ion and $c_{k0}$ of formaldehyde was used for the experiment. The complete equipment setup is described in Figure 19.

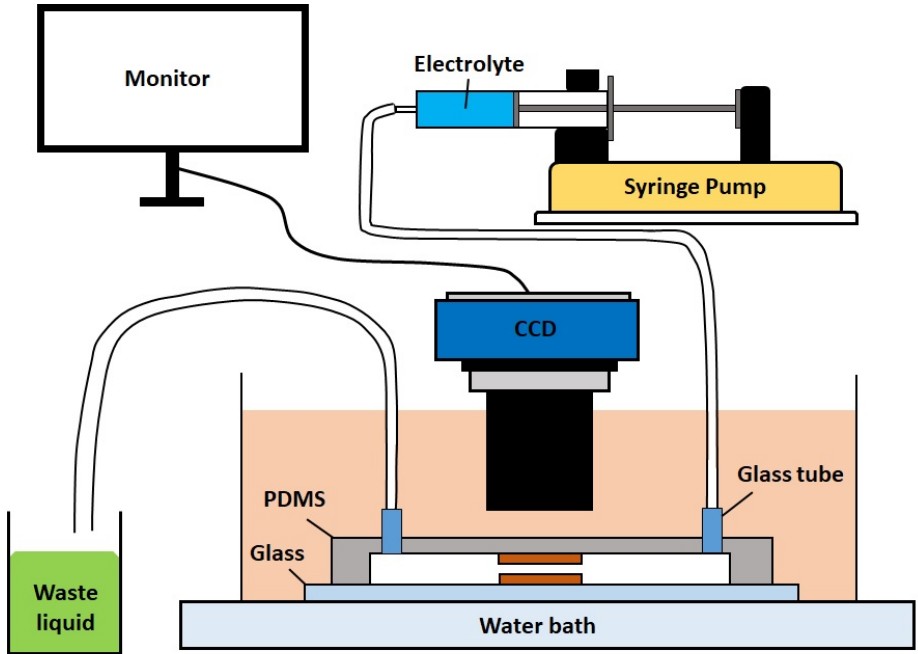

**Figure 19.** Electroless copper plating via using microfluidic system.

*6.2. Results*

Experimental results (see Figure 20) show that the bubbles are not only appearing on the copper plate, but also appearing on the top. In the video, one can see that there were several bubbles going to the top from the center or the bottom side of the channel. The region above the glass becomes darker with time. The simulation results (see Figure 13) qualitatively arrive at the same conclusion. The experiment indicates that the clustering of bubbles happens on both the top side and the bottom side of the channel. Second, the numerical simulation predicts that most bubbles are generated at an early stage and near the inlet. The experiment shows that the bubble generation is more exuberant near the inlet in comparison with other regions at $t = 20$. This observation coincides with that of Figures 2 and 3a. The region near the inlet at $t = 20$ is of the highest concentration of dissolving hydrogen gas. In addition, large bubbles were observed at the back end of the copper plates (i.e., region near (x,y) = (6,0) corresponding to Figure 4), which is also the case in Figure 3b.

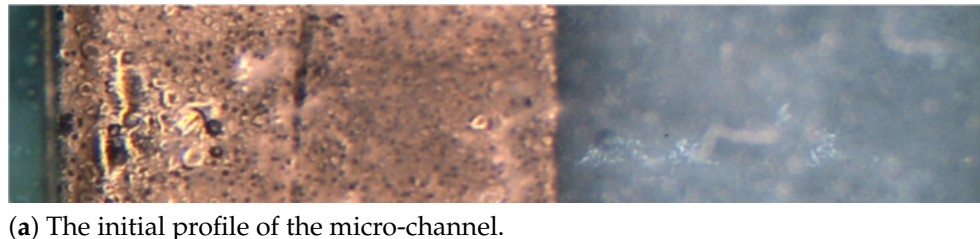

(**a**) The initial profile of the micro-channel.

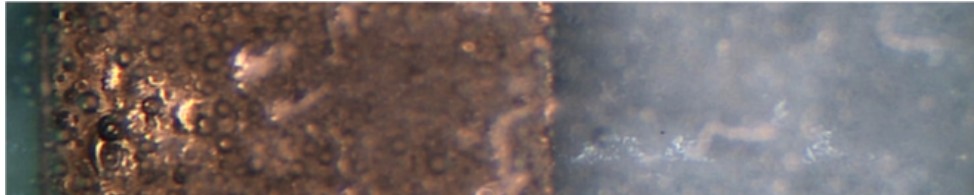

(**b**) Micro-channel at $t = 20$ s.

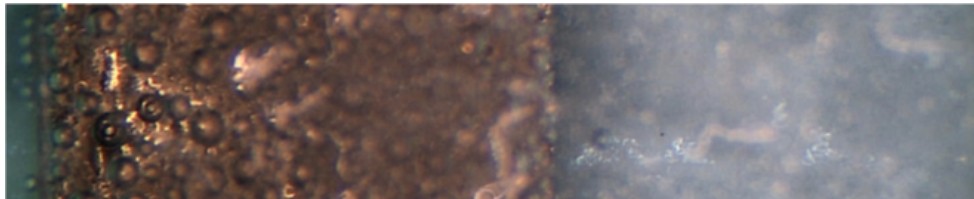

(**c**) Micro-channel at $t = 40$ s.

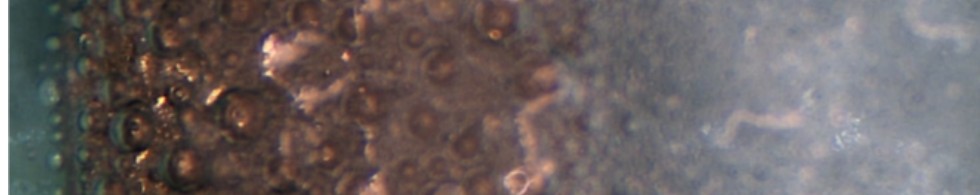

(**d**) Micro-channel at $t = 100$ s.

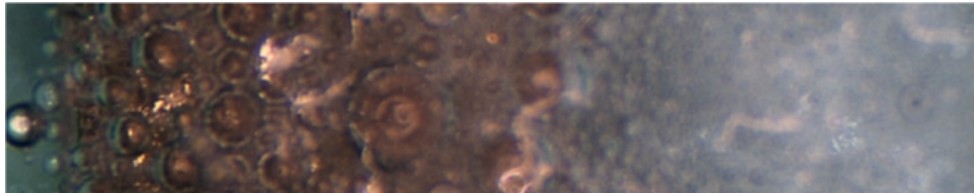

(**e**) Micro-channel at $t = 140$ s.

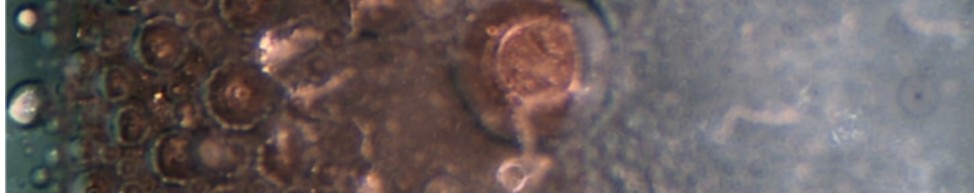

(**f**) Micro-channel at $t = 180$ s.

**Figure 20.** The pictures are taken from the top side and the region near the center between two 8 mm × 1 mm sheet glasses. The brown region is covered by the copper plate, where the surface reaction occurs.

### 6.3. Discussion

For an electroless plating process accompanying gas generation, the bubble distribution with respect to time, in the micro-channel, is the most important index for evaluating the quality of deposition. To measure it quantitatively, a high-quality optical system installed in the micro-channel is indispensable. For example, several types of fiber optical probes have been used to measure the particle (or bubble) size and distribution in a channel

flow (or micro-channel flow) [53–56]. However, such an optical system is difficult to install in our case because there is no appropriate place to set up the light source and the detector in the micro-channel. The signal interference caused by the copper plate or glued gel on two sides is almost inevitable.

## 7. Conclusions

The numerical simulation of electroless plating is difficult for two reasons: multi-phase modeling and nonlinearities. We have proposed a phase averaged liquid–gas two-fluid-velocity/one-pressure system combined with phase densities and chemical concentration equations. The nonlinearities being similar to those of the Navier–Stokes equations, we have used a semi-Eulerian time discretization leading to a generalized Stokes operator for the two-velocity/one-pressure system; the inf-sup saddle point theorem has lead to a proof of stability and well-posedness of the discretized system by the Hood–Taylor finite element method. The two-phase flow model is compatible with single phase models when the volume fraction of gas and the concentration of the gas in the liquid phase are set to zero. The model is also compatible with the one-dimensional model proposed in [8]. The numerical results confirm the robustness of the method. To validate the model, a real-life experiment has been performed. The numerical results agree qualitatively with the experiment for the repartition of bubbles near the plating boundary. We believe that in the future the computer code will be used to design industrial and experimental systems. However, as to the measurement of the deposition rate, It takes at least one hour to obtain an observable thickness of plating. In this case, bubbles have accumulated everywhere in the micro-channel and there is ground for an extension of the present code with a level set or phase field model which tracks the main liquid to gas interface. To establish a mathematical model suitable for a larger time simulation is left as future work.

**Author Contributions:** Conceptualization, P.-Y.W., O.P.; methodology, P.-Y.W., O.P.; software, P.-Y.W.; validation, P.-Y.W., O.P., P.-S.S., C.R.K.; formal analysis, P.-Y.W., O.P.; investigation, P.-Y.W., P.-S.S.; resources, O.P., C.R.K.; data curation, P.-Y.W., P.-S.S.; Writing—original draft preparation, P.-Y.W.; Writing—review and editing, O.P.; visualization, P.-Y.W., P.-S.S.; funding acquisition, C.R.K. All authors have read and agreed to the published version of the manuscript.

**Funding:** The experimental section of this research was financially supported by the Ministry of Science and Technology, Taiwan, under grant No. MOST 110-2622-E-002-016-CC1, the Ministry of Education, Taiwan. The authors are also grateful for the technical help provided by Taiwan Uyemura Co., Ltd.

**Data Availability Statement:** Not applicable.

**Acknowledgments:** We appreciate Han-Tang Hung provided several suggestions in terms of the experimental design. We are also grateful to Tony Wen-Hann Sheu for having initiated this work.

**Conflicts of Interest:** The authors declare no conflict of interest.

## Appendix A. Estimation of the Interfacial Terms

Let $V_0$ be a local volume to be observed which is occupied by gas and liquid. In a liquid–gas two-phase system, we have $A_l = A_g$ and further $\rho_l(w_l - u_l) \cdot n_l = -\rho_g(w_g - u_g) \cdot n_g$ on the interface. If the size of each single bubble in the electrolyte is small enough, then we can assume that the bubbles are spherical. Assuming that there is a typical radius for all bubbles $R_B > 0$ such that $1/R_B^2$ is the average of $1/R^2$ among all bubbles in the system, the growth rate of bubbles governed by the local mass loss prescribed by Equation (3) can be computed by the relation

$$4\pi R_B^2 N_q \frac{dR}{dt} = \int_{V_0} \frac{\dot{S}_g}{\rho_g} dV,$$ (A1)

where $N_q$ is the amount of bubbles in a local volume $V_0$. Therefore, we have the following formulae on $A_g$ and $A_l$, respectively

$$(\boldsymbol{u}_g - \boldsymbol{w}_g) \cdot \boldsymbol{n}_g = -\frac{1}{4\pi N_q R^2} \int_{V_0} \frac{\dot{S}_g}{\rho_g} dV \tag{A2}$$

$$(\boldsymbol{u}_l - \boldsymbol{w}_l) \cdot \boldsymbol{n}_l = \frac{1}{4\pi N_q R^2} \int_{V_0} \frac{\dot{S}_g}{\rho_l} dV. \tag{A3}$$

The quantity $R_B$ is useful when the fluid velocity is large enough so that each bubble will not stay at the observed physical domain, because every bubble has not been far from the state that is just after nucleation.

Given a small cube $V_0$ of size $|V_0| = d \times d \times d$ and a typical radius $R_B$, the ratio of its surface area and volume is $\frac{4\pi N_q R_B^2}{d^3}$, where $N_q$ can be estimated by

$$N_q = \frac{r_g d^3}{\frac{4}{3}\pi R_B^3} \tag{A4}$$

Therefore, if $d$ is small enough so that the physical quantities in $\boldsymbol{F}_\alpha$ defined in Section 2.3 can be assumed uniform, then we have the approximation

$$\boldsymbol{F}_l \approx \left(\frac{4\pi N_q R_B^2}{d^3}\right) \cdot \rho_l \cdot \left(-\frac{d^3 \dot{S}_g}{4\pi N_q R_B^2 \rho_l}\right) \boldsymbol{u}_l = -\dot{S}_g \boldsymbol{u}_l = \dot{S}_l \boldsymbol{u}_l. \tag{A5}$$

Similarly,

$$\boldsymbol{F}_g \approx \dot{S}_g \boldsymbol{u}_g \tag{A6}$$

The same approximation can be applied to $G_j$ occurring at (5) and (6):

$$G_j \approx \dot{S}_l c_j, \quad j = s, k, \quad G_g \approx \dot{S}_l c_g - M_g K \rho_l r_l (c_g - c_{sat})^+ \tag{A7}$$

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
