# Peer review of "Numerical Analysis of an Electroless Plating Problem in Gas–Liquid Two-Phase Flow"

_fluids, doi:10.3390/fluids6110371_

Round 1

Reviewer 1 Report

The text of the manuscript corresponds to the subject of the journal Fluids. In general, the manuscript is made at a high level, but there are a couple of comments:

1. I believe that all the same, the order of presetting for readers will be better if first the experiment, and then the numerical simulation. Because it is not clear what you are comparing when you are looking for an error.

2. Need more discussion about the results. Please explain every point. Conclusions should be presented more clearly.

Author Response

See attached pdf

Reviewer 2 Report

The authors have produced a novel numerical analysis study of a challenging problem -- electroless plating in a microchannel / microfluidic contacting pattern, where hydrogen bubbles are known to cause operational difficulties.  The analysis is by construction unique in the literature for this problem.  It is correct in all details that I could verify, and self consistent in approximation theory by testing for convergence and, to some extent, grid resolution.   Qualitatively, consistency with experimental observations provides some additional credence.   It is well known that the laminar Navier-Stokes equations are an excellent approximation in microchannel flows, with the error less than about 3% as found by Bandulasena et al. (2010).  Indeed, even the fidelity of 2-D approximations is well supported by 3-D stereoscopic PIV measurements (Bown et al., 2006).

So this paper should be recommended for publication as it is in scope and succeeds on its own terms.

I have two caveats.

  1. The two fluid approach is well developed for particles as the dispersed phase by Tsuji and co-workers.   It is implemented for bubbles as the dispersed phase by the commercial software package Comsol Multiphysics as the "bubbly flow model" and the "laminar bubbly flow model".   In fact. Comsol Multiphysics allows for using moving meshes, by the Arbitrary Lagrange-Eulerian formulation (built into the base unit of CMP), which is easy to apply to moving boundary conditions.   It is also fairly straightforward for dealing with receding or growing surfaces uniformly by simply using a coordinate mapping from the vertically-shrinking reference frame to a fixed frame.  Microbubble dynamics have been implemented in an airlift loop bioreactor by Al-Mashhadani et al. (2015) using the laminar bubbly flow model.  Hence, to a very great extent, the authors are "re-inventing the wheel" and should put their work in context of the literature, where they have stated that their approach is wholly original.  With the free surface treatment coupled to the two phase flow, the free surface component is a modest addition to the already existing two fluid approach.
  2. The authors state that it is impossible to measure the phase fraction in their system.   Actually, there are three well established approaches to inferring the bubble size distribution, from which the phase fraction can be computed with some geometry and arithmetic.   So certainly not impossible, but just impractical, as the off-the-shelf acoustic, optical, and laser diffraction approaches are not adapted to microfluidics scales / volumes of liquids.  See Desai et al. (2019).

Given the common denominator author in these references, and that literally wrote the book on Comsol Multiphysics, I cannot hide behind anonymity.  I cannot either require that more than one reference for which I am co-author be cited (MPDI policy).  Al-Mashhadani et al. (2015), however, is the closest to the modelling methodology adopted in this paper, so should be cited.   Comsol's documentation does not bear my name -- I am just one of the earliest adopters, but have no axe to grind.   Indeed, Ansys Fluent is a spinout of my department, so I am clearly independent!

So I am recommending for publication if the above issues can be adequately addressed.

H. C. H. Bandulasena,  W. B. Zimmerman and  J. M. Rees, "Rheometry of non-Newtonian polymer solution using     microchannel pressure driven flow", Applied Rheology, 20(5): U2-U9, 55608, 2010.

M R Bown, J M MacInnes, R W K Allen and W B J Zimmerman, “ Three-dimensional, three-component   velocity measurements using stereoscopic micro-PIV and PTV”, Meas. Sci. Technol. 17 2175-2185, 2006.

Al-Mashhadani MKH, Wilkinson SJ, Zimmerman WB, Airlift bioreactor for biological applications with microbubble   mediated transport processes, Chemical Engineering Science 137:243–253, 2015.

Desai P, Ng W, Hines M, Riaz Y, Tesar V, Zimmerman W, Comparison of Bubble Size Distributions Inferred from Acoustic, Optical Visualisation, and Laser Diffraction, Colloids and Interfaces (2019) 3(4) 65.

Author Response

See attached PDF
